



# Volcanic plume height during the 2021 Tajogaite eruption (La Palma) from two complementary monitoring methods. Implications for satellite-based products

África Barreto[1,2], Francisco Quirós[4], Omaira E. García[1], Jorge Pereda-de-Pablo[4], Daniel González-Fernández[2,3], Andrés Bedoya-Velásquez[5], Michael Sicard[6,7,8], Carmen Córdoba-Jabonero[9], Marco Iarlori[10], Vincenzo Rizi[10], Nickolay Krotkov[11], Simon Carn[11], Reijo Roininen[12], Antonio J. Molina-Arias[4], A. Fernando Almansa[13,1], Óscar Álvarez-Losada[14,1], Carla Aramo[15], Juan José Bustos[1], Romain Ceolato[5], Adolfo Comerón[6], Alicia Felpeto[4], Rosa D. García[14,1], Pablo González-Sicilia[14,1], Yenny González[13,1], Pascal Hedelt[16], Miguel Hernández[17], María-Ángeles López-Cayuela[9], Diego Loyola[16], Stavros Meletlidis[4], Constantino Muñoz-Porcar[6], Ermanno Pietropaolo[18], Ramón Ramos[1], Alejandro Rodríguez-Gómez[6], Roberto Román[2,3], Pedro M. Romero-Campos[1], Martin Stuefer[11], Carlos Toledano[2,3], and Elsworth Welton[19]

[1]Izaña Atmospheric Research Center (IARC), State Meteorological Agency of Spain (AEMET), Spain
[2]Group of Atmospheric Optics (GOA-UVa), Universidad de Valladolid, 47011, Valladolid, Spain
[3]Laboratory of Disruptive Interdisciplinary Science (LaDIS), Valladolid University, Valladolid, Spain
[4]Observatorio Geofísico Central, Instituto Geográfico Nacional, Madrid, 28014, Spain
[5]ONERA, The French Aerospace Lab, Universite de Toulouse, FR 31055 Toulouse, France
[6]CommSensLab, Dept. of Signal Theory and Communications, Universitat Politècnica de Catalunya (UPC), 08034-Barcelona, Spain
[7]Ciències i Tecnologies de l'Espai-Centre de Recerca de l'Aeronàutica i de l'Espai/Institut d'Estudis Espacials de Catalunya (CTE-CRAE/IEEC), Universitat Politècnica de Catalunya (UPC), 08034-Barcelona, Spain
[8]Laboratoire de l'Atmosphère et des Cyclones, Université de La Réunion, Saint Denis, 97744, France
[9]Instituto Nacional de Técnica Aeroespacial (INTA), Atmospheric Research and Instrumentation Branch, Ctra. Ajalvir, km.4, Torrejón de Ardoz, 28850-Madrid, Spain
[10]INFN-GSGC L'Aquila and CETEMPS-DSFC, Università degli Studi dell'Aquila, via Vetoio, 67100, L'Aquila, Italy
[11]NASA Goddard Space Flight Center, Greenbelt, 20771, EEUU
[12]Vaisala Oyj, 01670 Vantaa, Finland
[13]Scientific Department, CIMEL Electronique, 75011 Paris, France
[14]TRAGSATEC, 28037, Madrid, Spain
[15]INFN Napoli, Complesso Universitario Monte Sant'Angelo, Via Cintia - Ed. 6- Napoli, Italy
[16]German Aerospace Center (DLR), Remote Sensing Technology Institute, Oberpfaffenhofen, 82234, Weßling
[17]Regional Meteorological Center for Western Canary Islands, State Meteorological Agency of Spain (AEMET), Spain
[18]INFN-GSGC L'Aquila and DSFC, Università degli Studi dell'Aquila, via Vetoio, 67100, L'Aquila, Italy
[19]NASA, Goddard Space Flight Center, National Aeronautics and Space Administration, Greenbelt, MD 20771, USA

**Correspondence:** África Barreto (abarretov@aemet.es)

**Abstract.** Volcanic emissions from the Tajogaite volcano, located on the Cumbre Vieja edifice on the island of La Palma (Canary Islands, Spain), caused significant public health and aviation disruptions throughout the volcanic event (19 September – 13 December 2021, officially declared over on 25 December). The Instituto Geográfico Nacional (IGN), the authority responsible for volcano surveillance in Spain, implemented extensive operational monitoring to track volcanic activity and to provide a ro-





bust estimation of the volcanic plume height using a video-surveillance network. In parallel, the State Meteorological Agency of Spain (AEMET), in collaboration with other members of ACTRIS (Aerosol, Clouds, and Trace Gases Research Infrastructure) in Spain, in collaboration with other institutions, carried out an unprecedented instrumental deployment to assess the atmospheric composition impacts of this volcanic event. This effort included a network of aerosol profilers surrounding the volcano. A total of four profiling instruments were installed on La Palma: one MPL-4B lidar and three ceilometers. Additionally,

a pre-existing Raman lidar on the island contributed valuable data to this study.

In this study, the eruptive process was characterised in terms of the altitude of the dispersive volcanic plume ($h_d$), measured by both IGN and AEMET-ACTRIS, and the altitude of the eruptive column ($h_{ec}$), measured by IGN. Modulating factors such as seismicity and meteorological conditions were also analysed. The consistency between the two independent and complementary datasets ($h_{d,IGN}$ and $h_{d,AEMET}$) was assessed throughout the eruption (mean difference of 258.6 m).

Our results confirmed the existence of three distinct eruptive phases, encompassing a range of styles from Strombolian explosive to effusive activity. While these phases have been characterised in previous studies, the results of the present work provide complementary information and novel insights from a different scientific perspective, which may be of use in future volcanic crises and will be applied to operational surveillance during such events.

A subsequent comparison of $h_{d,AEMET}$ with the Cloud-Aerosol Lidar with Orthogonal Polarization (CALIOP) aerosol

layer height product ($ALH_{CALIOP}$) revealed a systematic underestimation by the satellite product, with a mean difference of 392.2 m.

Finally, the impact of using $h_{ec}$ in estimating $SO_2$ emissions from the NASA $MSVOLSO2L4$ satellite-based product was evaluated. When a fixed (standard) plume altitude of 8 km was used instead of the observed $h_{ec}$, the total $SO_2$ mass was significantly underestimated by an average of 56.2%, and by up to 84.7%. These findings underscore the importance of

accurately determining the volcanic plume height when deriving $SO_2$ emissions from satellite data.

## 1 Introduction

Anthropogenic aerosols, as one of the major drivers of climate change, have attracted significant scientific attention for decades. Natural aerosols, such as dust, sea salt, and volcanic particles, have also been thoroughly studied in recent decades, particularly

in relation to their radiative forcing (Boucher et al., 2013; Masson-Delmotte et al., 2021; Forster et al., 2021). Volcanic aerosols, both tropospheric and stratospheric, have a primary impact on atmospheric chemistry, climate, and the radiative budget. Moreover, they play a critical role in public health, civil aviation, the economy, and ecosystems (Carn et al., 2017; Karagulian et al., 2010; Kampouri et al., 2020; Aubry et al., 2021; Nogales et al., 2022; García et al., 2023).

Volcanoes primarily emit gaseous species, including water vapour ($H_2O$), carbon dioxide ($CO_2$), and sulfur dioxide ($SO_2$),

the latter being the most abundant gas released during volcanic activity (Gebauer et al., 2024). Primary aerosols can be directly



emitted at the vent, such as ash and sulfate aerosols (Karagulian et al., 2010). Secondary sulfate aerosols are formed through in-plume conversion of $SO_2$ into sulfuric acid droplets via gas-phase oxidation (Boichu et al., 2019; Kampouri et al., 2020), along with other sulfate-bearing compounds (Gebauer et al., 2024). The efficiency of this conversion process from primary to secondary aerosols is influenced by several factors and generally increases with temperature and relative humidity (Gebauer

et al., 2024, and references therein). These newly formed sulfate aerosols can subsequently grow via coagulation and condensation during downwind transport (Aubry et al., 2021), significantly affecting atmospheric chemistry on local scales. Their impacts include drastic changes in atmospheric chemistry, the degradation of air quality, and their role as cloud condensation nuclei (CCN) and ice-nucleating particles (INPs) (Pappalardo et al., 2004; Kampouri et al., 2020; Aubry et al., 2021; Gebauer et al., 2024). Taking into account that the lifetime of sulfate aerosols is significantly longer than that of primary volcanic emis-

sions (typically around 1.3 weeks in the troposphere and several years when injected into the stratosphere), secondary volcanic aerosols can be transported over long distances and may exert a global impact on the climate system (Pappalardo et al., 2004; Gebauer et al., 2024), such as increasing the Earth's albedo or contributing to ozone depletion in the stratosphere. The amount of incoming solar radiation scattered from these emitted atmospheric components, and therefore the radiative effect and impact of the volcanic aerosols, depend on the location of the volcanic layer, the size of the emitted particles (Marshall et al., 2020),

and also the nature of the volcanic aerosol. Although volcanic sulfates are expected to be non-absorbing fine-mode, spherical particles mainly scatterers of solar radiation, volcanic ash is observed to be irregular coarse-mode and more absorbing particles (Ansmann et al., 2012; Sellitto and Briole, 2015).

Tropospheric volcanic aerosols originating from smaller eruptions and sustained magmatic or hydrothermal degassing remain largely understudied, primarily due to the lack of detailed 'near-source' characterisations (Mather et al., 2003; Sellitto

et al., 2016; Sicard et al., 2022; Córdoba-Jabonero et al., 2023; Taquet et al., 2025). Despite their relatively modest scale, these emissions can have a significant impact on air quality, as they may be injected into either the planetary boundary layer or the free troposphere, depending on the interplay between plume height and atmospheric vertical structure. In addition to their role in air pollution, such aerosols may also influence cloud formation and radiative properties (Mather et al., 2003). Consequently, the atmospheric and climatic effects of weak volcanic activity are still poorly represented in current climate models (Sellitto

et al., 2016).

The eruptions of the Eyjafjallajökull volcano in Iceland during April and May 2010 provided the scientific community with an unprecedented opportunity to study, in near-real time, the role of volcanic aerosols in the climate system. The event also had a major impact on air traffic and the global economy, as the volcanic plume was transported over Europe (Ansmann et al., 2010, 2011, 2012; Sicard et al., 2012; Navas-Guzmán et al., 2013, and references therein). As noted by Ansmann et al. (2012),

this eruption triggered significant advancements in lidar technology and promoted its broader deployment by research and environmental institutions, given the suitability of lidar systems for detecting transported aerosol plumes at high altitudes, and their ability to assess aerosol type through depolarisation measurements.

Other ground-based techniques, such as video-surveillance systems, weather radars, infrasound, and lightning detection, have also been used to estimate volcanic plume top height (Lamb et al., 2015). Among these, video-surveillance systems offer



a particularly promising and easily automatable solution for near-real-time estimation of $h_{ec}$ during emergency events (Scollo et al., 2014; Kampouri et al., 2020; Felpeto et al., 2022).

The magnitude of $SO_2$ mass loadings (or emitted fluxes) and the height of the volcanic plume are key parameters for monitoring volcanic activity and estimating the radiative and climatic impacts of volcanic eruptions (Theys et al., 2013; Marshall et al., 2020; Fedkin et al., 2021). Sulfur emission rates during eruptions can be quantified through direct observations from

various platforms (Kremser et al., 2016), including satellite remote sensing and in situ measurements (Masson-Delmotte et al., 2021; Taquet et al., 2025). Satellite-based hyperspectral spectrometers operating in the ultraviolet (UV) have provided frequent and increasingly accurate observations of global $SO_2$ levels. These observations rely on retrieval algorithms that process backscattered radiance measurements to estimate both $SO_2$ mass loadings and plume height (Fedkin et al., 2021; Carn et al., 2016, 2017; Carn, 2022; Hedelt et al., 2025). According to Theys et al. (2013) and Carn (2022), the greatest source of un-

certainty in estimating $SO_2$ mass loadings in the lower troposphere arises from the limited a priori knowledge of volcanic plume altitude. $SO_2$ retrievals are more accurate for plumes located above the $SO_2$ cloud and snow/ice layers, with overall uncertainties typically ranging from 20–30% (Carn, 2022).

The 2021 volcanic eruption on La Palma (Canary Islands, Spain) served as a natural laboratory for studying the eruptive process and its impacts from multiple perspectives (e.g., Román et al., 2021; Sicard et al., 2022; Bedoya-Velásquez et al., 2022;

Carracedo et al., 2022; Nogales et al., 2022; García et al., 2023; Córdoba-Jabonero et al., 2023; Milford et al., 2023; Taquet et al., 2025). The evolution and long-range transport of the volcanic plume have been the subject of numerous studies, including its dispersion toward the Iberian Peninsula (Salgueiro et al., 2023), southern France (Bedoya-Velásquez et al., 2022), and Cape Verde (Gebauer et al., 2024). In this context, this study aims to present the unprecedented instrumental coverage deployed during the Tajogaite (Cumbre Vieja) eruption (19 September to 25 December 2021) by the Instituto Geográfico Nacional

(IGN), the Spanish State Meteorological Agency (AEMET), and other Spanish members of ACTRIS (Aerosol, Clouds and Trace Gases Research Infrastructure) to monitor the atmospheric impact of this rare event. As the institution responsible for volcano surveillance in Spain, IGN implemented extensive monitoring from the beginning of the eruption, including robust estimations of the volcanic plume height using a video-surveillance system. This quantification was incorporated into the PEVOLCA (Steering Committee of the Special Plan for Civil Protection and Attention to Emergencies due to Volcanic Risk)

reports (PEVOLCA, 2021), the VONA (Volcano Observatory Notice for Aviation) alerts, and the regular reports submitted to the Toulouse VAAC (Volcanic Ash Advisory Centre) during the crisis (VAAC, 2022; Felpeto et al., 2022). In parallel, AEMET and ACTRIS (hereafter AEMET-ACTRIS) established a network of aerosol profilers around the volcano to provide a complementary and reliable estimation of the dispersive plume altitude. A total of four profiling instruments were deployed on La Palma, in addition to one pre-existing instrument, all located within 15 km of the volcano. These included one Raman

lidar (ARCADE), one Micropulse Lidar (MPL), and three ceilometers (Vaisala CL51, Vaisala CL61, and Lufft CHM15k). The information obtained from these profilers was compared with the dispersive plume height ($h_d$) estimated by IGN's video-surveillance system and were also used in PEVOLCA Committees. This approach enabled the retrieval of the eruptive column height ($h_{ec}$) and the altitude of the dispersive plume ($h_d$).



The manuscript is organised as follows. Section 2 describes the main features of the volcanic eruption. Section 3 details the
IGN and AEMET-ACTRIS monitoring networks, as well as the auxiliary information used in this study. The main results are
presented in Section 4, including the comparison between the two monitoring networks (Section 4.1) and the description of the
volcanic event from volcanic plume altitudes and modulating factors (Section 4.2). The evaluation of satellite-based products
is included in Section 4: CALIPSO (Cloud-Aerosol Lidar and Infrared Pathfinder Satellite Observations) CALIOP (Cloud-
Aerosol Lidar with Orthogonal Polarization) aerosol height (Sections 4.3), and the estimation of $SO_2$ volcanic emissions using
multi-satellite UV-based observations (Section 4.4). Finally, Section 5 presents the summary and conclusions that are extracted
from this work.

## 2  Tajogaite 2021 volcanic eruption

On 19 September 2021, at the Cumbre Vieja volcanic edifice, an eruption began. The initial phase was a fissural eruption, and
over the 85 days of eruption duration, significant changes were detected in the activity. PEVOLCA Committee declared offi-
cially the end of the eruption on 25 December (which ended on 13 December from a seismic perspective) (PEVOLCA, 2021).
In September 2022, nearly one year after the eruption began, the volcano on Cumbre Vieja was officially named Tajogaite, a
traditional toponym of Guanche origin used by locals to refer to the upper part of the affected area.

This eruption is considered the most significant volcanic event in Europe over the past 75 years due to the substantial amount
of $SO_2$ released into the atmosphere, and the extensive damage caused by lava flows (Rodríguez et al., 2022). It is also regarded
as the longest-lasting historical eruption on the island. The eruption led to the evacuation of more than 7000 residents, with
severe consequences for public health and a profound impact on the island's economy. These effects persist, owing to the vast
lava fields (extending over an area of about 1219 ha), widespread destruction of infrastructure, homes, and farmland, and the
continued emission of volcanic gases in an area heavily reliant on tourism. Flight operations were also significantly disrupted:
26% of scheduled flights at La Palma Airport were cancelled, 34% due to airport closures caused by ash accumulation, and
the remaining 66% due to the presence of volcanic ash in the airspace (Benito et al., 2023). Initial estimates place the total
economic losses at over 1025 million USD (Benito et al., 2023).

Multiple summit vents emerged during the nearly three months of volcanic activity, exhibiting a range of eruptive styles—from
Strombolian explosions to effusive phases—resulting in violent ash-rich eruptions, lava fountains, gas emissions, and extensive
lava flows. The main volcanic edifice, a cinder cone that reached an altitude of 1121 m above sea level (a.s.l.), was formed in
the northwestern sector of the Cumbre Vieja rift (Carracedo et al., 2022; Felpeto et al., 2022; Romero et al., 2022; Benito et al.,
2023).

Although the eruption was predominantly characterised by Strombolian basaltic activity, different eruptive phases were
observed throughout the entire eruption period (Del Fresno et al., 2022; Milford et al., 2023; Benito et al., 2023). As noted by
Milford et al. (2023), these phases were distinguishable based on variations in surface and satellite-derived $SO_2$ emission rates.
According to their results, the initial phase of the eruption exhibited the highest $SO_2$ and ash emissions, with an accumulated
$SO_2$ output of 1.59 Mt. This was followed by a marked decline in emissions during a second phase, beginning around 7



November, with an accumulated $SO_2$ emission of 0.25 Mt. According to Benito et al. (2023), nine different eruptive phases can be identified in terms of seismological data (epicenters, hypocenters and magnitude/depth of earthquakes). Del Fresno et al. (2022) attended to the variation in the number of events with magnitudes between 2–3 mbLg, as well as changes in the

Real-time Seismic Amplitude Measurement (RSAM) values using the seismic pattern recorded by the IGN monitoring network (https://www.ign.es/web/ign/portal/sis-catalogo-terremotos). Such variations can be interpreted as upwards migration of fresh, $SO_2$-rich magma batches and related degassing processes through the eruptive vents (Palma et al., 2008).

## 3  Instruments and Methods

### 3.1  IGN video-surveillance monitoring network

The IGN, as the competent institution for volcano surveillance in Spain, monitored volcanic events in terms of seismic activity, ground deformation, geochemical and gravimetric parameters, and volcanic plume behaviour (Lopez et al., 2022). In this context, a monitoring network based on video-surveillance cameras was implemented to track the altitude of the volcanic plume. This methodology provided the quantitative characterisation of the volcanic plume required by the PEVOLCA Committees during the crisis. The results were also incorporated into the VONA alerts and the regular reports submitted to the Toulouse

VAAC (VAAC, 2022; Felpeto et al., 2022).

The monitoring network consisted of calibrated webcams that provided scaled images of the volcanic plume. A pre-existing webcam from the E.U. EELabs project (https://www.eelabs.eu/), coordinated by the Instituto de Astrofísica de Canarias (IAC, https://www.iac.es/) and located at the Roque de Los Muchachos Observatory (28.76°N, 17.88°W, 2423 m a.s.l.; approximately 16 km north of the eruption site), was selected as the reference calibrated webcam. This camera had to meet several criteria

to be suitable for measuring the altitude of the volcanic plume. A favourable location was essential: the camera needed to be situated at a high altitude above sea level and far enough from the eruption to capture the full extent of the plume, while minimising the effect of plume inclination and the obstruction by volcanic material. The camera's position and internal settings must remain unchanged during the eruptive period. Furthermore, it was necessary to ensure continuous data recording and easy access to the image archive. The IAC camera met all these requirements and had already been recording prior to the eruption.

Additionally, three more webcams were installed by IGN personnel at different locations surrounding the volcanic vent: Time (28.66°N, 17.94°W), Los Llanos de Aridane (28.67°N, 17.92°W), and Fuencaliente (28.50°N, 17.85°W). These cameras were used to measure the height of the volcanic plume from different angles, allowing for a better understanding of the influence of wind on plume dispersion.

In order to achieve the objective of detecting the altitude of the volcanic plume, it is essential to determine the scale of the

images based on a specific distance, particularly the distance from the camera to the volcanic vent. By employing geodetic techniques such as trigonometric levelling, we obtained different altitudes above sea level along the eruptive column at specific moments and selected locations. Once these altitudes were retrieved, they were compared with the corresponding points identified in the images.





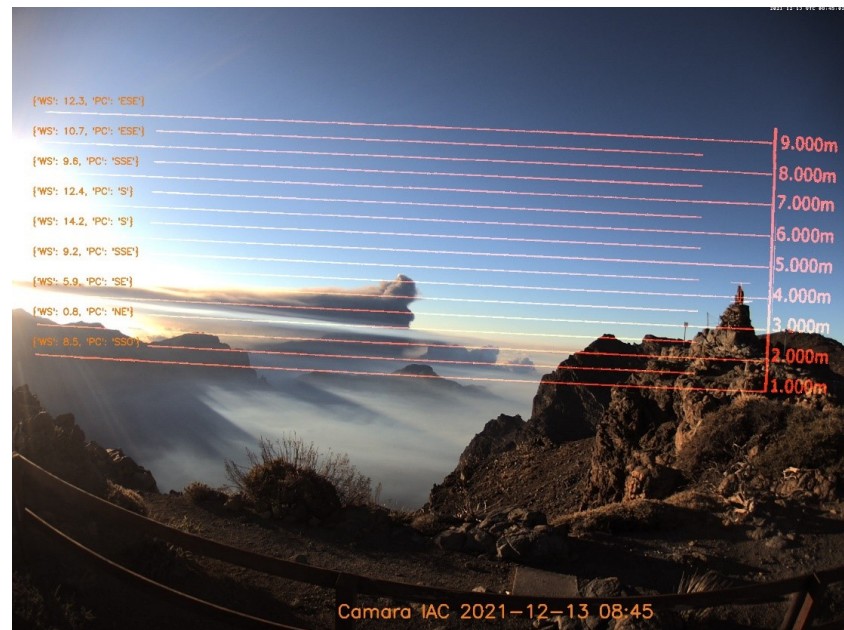

**Figure 1.** Calibrated image taken from the IGN-IAC webcam on 13-12-2021 with grid meteorological information from AEMET (wind speed, WS in m s$^{-1}$) and wind direction (in terms of Cardinal Points, PC).

To perform direct measurements on the images, a calibrated scale needed to be established (see Fig. 1). The calibration

process involved multiple simultaneous measurements of the same reference points using both image analysis and geodetic methods. A trigonometrical station, located approximately 30 meters from the camera, was used for this purpose. The resulting scale was used to generate a grid that was automatically superimposed on the images, enabling real-time measurements of the volcanic plume height on demand. This grid also included wind direction and speed at different altitudes, as presented in Fig. 1, which was crucial for accurate estimation of the column height.

This procedure was also applied to the camera installed at the Time station. For the other two webcams, located in Los Llanos de Aridane and Fuencaliente, calibration was performed by comparing their images to those from the calibrated reference camera at Roque de Los Muchachos (RMO), taken at the same moments. Unfortunately, the Fuencaliente camera was installed near the end of the eruption and provided limited useful data.

This technique has its limitations. Since it relies on visual observations from cameras, it is generally not effective during

nighttime. However, during the most intense phases of an eruption, it was possible to observe the eruptive column even at night. On days with heavy cloud cover or fog, identifying the top of the eruptive column proved challenging. Furthermore, during certain stages of the eruption, particularly in the final weeks when ash emissions were weak and the volcano displayed a pulsating behaviour, producing discrete ash puffs every few seconds, the visibility was significantly compromised.





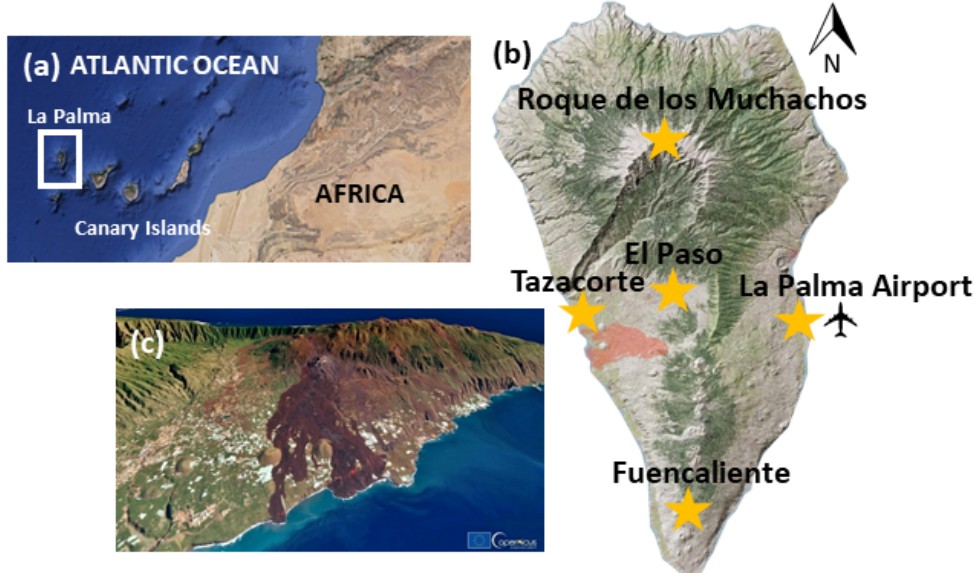

**Figure 2.** (a) Location of LA Palma and the Canary Islands in the Atlantic Ocean, (b) location of the five AEMET-ACTRIS stations (orange stars) deployed in La Palma and (c) 3D visualisation generated using the image acquired by one of the Copernicus Sentinel-2 satellites on 3 January 2022. The red spot in (b) marks the location of the volcanic vents. Credits on the map: © Google Earth 2023, GRAFCAN 2023 and European Union, Copernicus Sentinel-2 imagery 2022

Using this methodology, the height reached by the eruptive column ($h_{ec}$) and the dispersive plume ($h_d$) can be calculated, the latter referring to the altitude attained by the volcanic plume as it is transported and modified by thermodynamic processes, including atmospheric stratification, wind direction, and turbulence.

The IGN altitude database ($h_{ec,\text{IGN}}$ and $h_{d,\text{IGN}}$) consists of 344 individual altitude measurements corresponding to the eruptive column and dispersive volcanic plume heights recorded between 20 September and 13 December 2021. Each observation was performed at the time when a significant change in the height of the eruptive column was detected. Therefore, it can be assumed that both series remained relatively stable between consecutive measurements.

### 3.2 AEMET-ACTRIS profiling monitoring network

Four stations were strategically deployed in record time around the Tajogaite-Cumbre Vieja volcano by AEMET-ACTRIS members in Spain in collaboration with other institutions (Fig. 2). These stations, equipped with various types of cloud and aerosol profilers, enabled continuous tracking of the volcanic plume under variable wind conditions throughout the entire eruption, as well as providing ground-level air quality information and plume height for PEVOLCA reports. In addition to these, one station that was already installed in La Palma prior to the eruption was also added to the profiling network.

The characteristics of the five sites, as shown in Fig. 2b, are as follows:



- Roque de los Muchachos (RMO, 28.75°N, 17.88°W, 2423 m a.s.l.): This high-altitude observatory, part of the Instituto de Astrofísica de Canarias (IAC), is dedicated to high-quality astrophysical observations. It hosts telescopes and other astronomical instruments from nineteen countries, in addition to an AERONET (Aerosol Robotic Network, https://aeronet.gsfc.nasa.gov) station, which provides valuable aerosol properties information for this site. RMO is located 16.5 km from the eruptive vents.

- Tazacorte (TAZ, 28.64°N, 17.93°W, 140 m a.s.l.): Situated approximately 4 km from the eruptive vents in the town of Tazacorte (population ∼ 4500) on the eastern flank of La Palma, this station offers an excellent opportunity to study the aerosol layer near ground level when the predominant trade winds favor the dispersion of the volcanic plume westward.

- Fuencaliente (FUE, 28.49°N, 17.85°W, 630 m a.s.l.): Located on the southern side of La Palma, with a population of fewer than 2000 people, this station is about 10 km from the volcanic vents. Positioned in a rural background site, it provides a valuable opportunity to detect the volcanic plume under the prevailing trade wind regime.

- La Palma Airport (LPA, 28.62°N, 17.75°W, 56 m a.s.l.): This station is crucial for providing civil aviation with reliable information regarding the potential presence of volcanic ash, which can severely impact aircraft fuselages, sensors, and engines.

- El Paso (EPA, 28.65°N, 17.87°W, 700 m): Located in the central-northern part of La Palma, the municipality of El Paso has a population of approximately 7700. The station here is the closest to the volcano, at about 3 km from the volcanic vents, allowing for early detection of the volcanic plume during its initial stages. However, it also impacts the instruments in terms of the quality of the measurement, mainly in the most intense stages of the eruption.

By combining information from these five sites, a single $h_{d,\mathrm{AEMET}}$ time series was constructed, given that the proximity of the stations to the volcano and to each other ensures that no significant dispersive processes occur either horizontally or vertically, nor are intra-plume chemical reactions expected. This assumption is supported by previous findings on intra-plume interactions (e.g., Taquet et al., 2025). A total of 137 altitudes of the dispersive column ($h_{d,AEMET}$) have been included in the comparison analysis, coincident with those values measured by IGN. $h_{d,AEMET}$ observations from these five profilers have been selected for the comparison with IGN's database as those lying in a time interval of ± 5 min around IGN observation time. Taking into account the inherent problems in retrieving $h_{d,AEMET}$ from this on-site profiler network (signal blocking by the dense volcanic plume, high signal-to-noise ratio (SNR), cloud contamination, or technical problems), we have considered $h_{d,AEMET}$ also valid for the intercomparison provided that they lie between two IGN observations, since $h_{d,IGN}$ is not expected to change appreciably in the interval between two consecutive IGN measurements (see Sect. 3.1).

Detailed information on each system that is part of the network will be presented below, including the different methodologies used in this monitoring network. The approaches used in this study, that is, the Gradient (Flamant et al., 1997) and the Wavelet Covariance Transform (Brooks, 2003; Baars et al., 2008) methods, were applied to all AEMET-ACTRIS profilers except the RMO station. These methodologies can be considered equivalent and reliable according to Comeron et al. (2013).





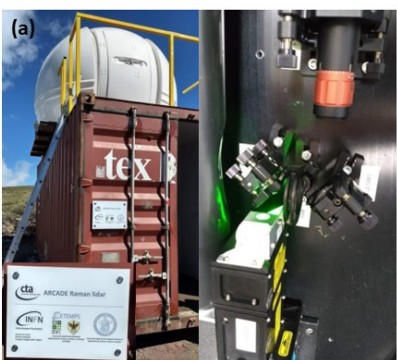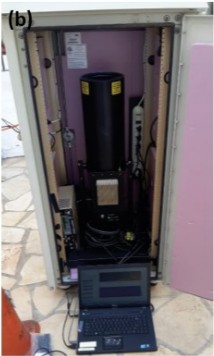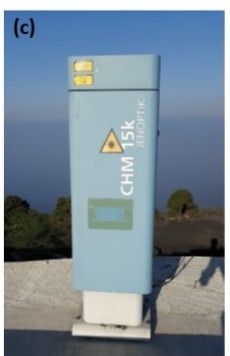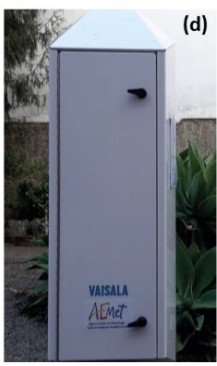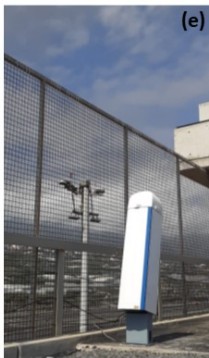

**Figure 3.** View of the different instruments belonging to the AEMET-ACTRIS profiling network: (a) ARCADE Raman lidar at Roque de los Muchachos, (b) MPL-4B at Tazacorte, (c) CHM15k at Fuencaliente, (d) CL61 at El Paso and (e) CL51 at La Palma Airport.

### 3.2.1 The ARCADE Raman Lidar at Roque de los Muchachos

This was the only existing profiler in La Palma at the time of the eruption, providing the only information about plume height until the next profiler at Fuencaliente was installed, 11 days after the eruption began. ARCADE is a Raman Lidar system (Fig. 3a) installed at the Cherenkov Telescope Array Observatory (CTAO) at RMO in October 2018, with the aim of providing nightly and seasonal aerosol attenuation profiles of UV light measured for the CTA two times a day, at sunrise and sunset, in automatic and unattended mode. Since the production of Cherenkov light depends on the molecular profile, aerosol profiles are required to estimate the attenuation and propagation in CTA observations (Iarlori et al., 2019). ARCADE is a joint project between CTA members from the National Institute for Nuclear Physics (INFN) and the University of L'Aquila.

This system is designed to measure elastic and Raman backscattered photons using a 20 cm diameter Newtonian telescope and a Nd:YAG laser source operating at 355 nm. The laser delivers pulses with a maximum energy of 5–6 mJ and a variable repetition rate ranging from 1 to 100 Hz (Valore et al., 2017; Iarlori et al., 2019). A second Raman channel for water vapour was added to retrieve atmospheric water vapour profiles; however, due to the COVID-19 outbreak, the Raman channels have been offline since 2021. The system was dismantled in October 2023 for future deployment at CTAO-South.

$h_{d,AEMET}$ was qualitatively estimated with this instrument from 19 September to 27 October at two fixed times per day (06:00 and 19:00 UTC), resulting in a total of 18 $h_{d,AEMET}$ values retrieved from this site. The estimation was based on visual inspection of the aerosol backscatter profiles, when available. Although the approach used at RMO is qualitative, the plume altitude measurements provided by this instrument have been included in this study, as they represent the only available information on the volcanic plume during the initial phase of the eruption and were the sole profiler data from the northern part of La Palma until the CL61 at El Paso station became operational on 25 October. This information proved to be extremely valuable during the eruptive process, with some of the ARCADE-derived altitudes reportedly used by PEVOLCA. From 27




October 2021 onwards, a technical issue prevented the ARCADE team from continuing plume monitoring, and the instrument's operation was definitively discontinued.

### 3.2.2 MPL-4B micropulse lidar at Tazacorte

The MPL-4B is a micropulse lidar (Campbell et al., 2002; Welton and Campbell, 2002) installed at Tazacorte station on 15 October 2021 (Fig. 3b), 25 days after the eruption started. This instrument is jointly owned by the Universitat Politècnica
de Catalunya (UPC) and the Instituto Nacional de Técnica Aeroespacial (INTA). MPL operates continuously (24/7) with a relatively high frequency (2500 Hz) and low-energy ($\sim 7\ \mu$J) using a Nd:YLF laser centred at 532 nm. As a part of NASA/M-PLNET (https://mplnet.gsfc.nasa.gov, last access: 15 March 2025), this system provides vertical profiles of clouds and aerosols with a 1-min temporal resolution and 75 m of vertical resolution. MPL-4B also includes polarisation capabilities, which are particularly useful to distinguish between spherical and non-spherical particles (Flynn et al., 2007).

The methodology used to retrieve $h_{d,AEMET}$ from this instrument is based on in-house algorithms, as described in Córdoba-Jabonero et al. (2018); Sicard et al. (2022); Córdoba-Jabonero et al. (2023). In our case, the Range-Square-Corrected lidar Signal (RSCS) was integrated over a $\pm 5$-minute interval around the IGN observation time in order to retrieve the height corresponding to the peak of the uppermost volcanic plume. A reference RSCS value at 8 km was used, and a simple threshold method (Melfi et al., 1985) was applied to the RSCS profiles (Measures, 1984) to detect cases of signal blocking or strong
attenuation. Further details on the algorithms employed in this study can be found in Sicard et al. (2022); Córdoba-Jabonero et al. (2023).

A total of 24 altitudes retrieved with this MPL-4B instrument constitute the final $h_{d,AEMET}$ data series.

### 3.2.3 Lufft CHM15k ceilometer at Fuencaliente

This Lufft CHM15k ceilometer (Jenoptik, 2013), belonging to the Group of Atmospheric Optics of the University of Valladolid
(GOA-UVa), is part of the Iberian Ceilometer Network (Cazorla et al., 2017). The CHM15k operates continuously (24/7) with a pulse energy of 8.4 $\mu$J and a repetition frequency of 5-7 kHz (Fig. 3c). The pulsed Nd:YAG laser emits at 1064 nm, and the backscattered signal is collected using a telescope with a field of view of 0.45 mrad and a vertical resolution of 15 m. The use of near-infrared (nIR) laser emission minimises uncertainty due to avoiding water vapour absorption, compared to ceilometers operating at lower wavelengths. According to Heese et al. (2010), the full overlap of the system is reached at approximately
1500 m a.s.l. Other studies (Cazorla et al., 2017; Román et al., 2017; Kotthaus et al., 2016) report that the maximum detection height of this system is 15360 m a.s.l., and that the overlap-corrected signal begins at 80 m a.s.l. More technical details about the CHM15k can be found in Jenoptik (2013).

This ceilometer, installed at the Fuencaliente station on 30 September, provided RSCS profiles to this study from all products available in the CHM15k Nimbus internal software. These profiles were integrated over a $\pm 5$-minute interval around the IGN
observation time. As in the MPL procedure, 8 km was considered the maximum altitude with significant aerosol influence. The gradient method (Flamant et al., 1997) was used to detect the uppermost aerosol layer based on sharp gradients in the



backscattered signal, which are sensitive to aerosol concentration. An empirical SNR threshold of 3 was applied to limit the maximum height at which data are considered reliable (Morille et al., 2007).

Using this methodology, a total of 63 altitudes were included in the final $h_{d,AEMET}$ dataset.

### 3.2.4 Vaisala CL51 ceilometer at La Palma Airport

The CL51 ceilometer (Fig. 3d), belonging to AEMET, operates continuously (24/7), using a pulsed laser emitter centred at 910 nm to measure backscattered radiation via a telescope in a coaxial configuration with an avalanche photodiode detector. This configuration allows the system to achieve an overlap height as low as 50 m, according to the manufacturer (Bedoya-Velásquez et al., 2022). The spatial and temporal resolutions of the CL51 are 10 m and 15 s, respectively. The capability of this instrument for aerosol retrieval has been validated in several studies (Bedoya-Velásquez et al., 2021, and references therein), and was specifically applied to the La Palma volcanic eruption in Bedoya-Velásquez et al. (2022). The methodology used to retrieve the temporal evolution of attenuated backscatter profiles is described in the latter study and is based on the Continuous Wavelet Covariance Transform (WCT) method (Brooks, 2003; Baars et al., 2008; Morille et al., 2007).

This instrument was installed at La Palma Airport on 7 October and contributed a total of 26 altitudes to the final $h_{d,AEMET}$ dataset.

### 3.2.5 Vaisala CL61 lidar ceilometer at El Paso

The Vaisala CL61 ceilometer (Fig. 3e) is a novel, high-performance instrument capable of depolarisation measurements, designed for unattended 24/7 operation under all weather conditions. It uses a single avalanche photodiode detector along a coaxial optical path. The CL61 operates at a wavelength of 910.5 nm, with a vertical resolution of 4.8 m, a temporal resolution of up to 5 s, and a maximum range of up to 15400 m. Its patented optical design enables the use of a narrow spectral bandwidth, which minimises water vapour absorption and ensures wavelength stability across varying temperatures.

The transmitter operates at a pulsing frequency of 9.5 kHz, with an average laser power of 40 mW and a pulse length of 160 ns (Full Width at Half Maximum, FWHM). The CL61 also incorporates an enhanced single-lens optical system with a coaxial configuration for both transmitted and received signals, featuring individual overlap functions calibrated for each instrument. This optimised overlap function enables the reliable detection of atmospheric layers at low altitudes.

The attenuated backscatter profiles are pre-calibrated for liquid water clouds at the Vaisala facilities, following the methodology described in O'Connor et al. (2004). Since the CL61 uses the same receiver module for both cross-polarised (x-pol) and parallel-polarised (p-pol) signals, no additional receiver sensitivity calibration is required.

A similar methodology to that described in the previous section for the CL51 was applied to the CL61 to retrieve attenuated backscatter profiles, following Bedoya-Velásquez et al. (2022).

Six altitude levels retrieved by the CL61 at the El Paso station were included in the final $h_{d,AEMET}$ dataset, contributing low statistical significance to the data from this station.



### 3.3 Auxiliary information

#### 3.3.1 Raw real-time Seismic Amplitude Measurement (RSAM)

Seismic records and ground deformation analyses are critically important for monitoring eruptive events (Carracedo et al., 2022). The Real-time Seismic Amplitude Measurement (RSAM), based on the vertical component of broadband seismic data, is widely recognised as a reliable indicator of magma transport from depth to the surface vent (Endo and Murray, 1991; Bartolini et al., 2018).

In this study, RSAM data were extracted from the IGN permanent seismic network at the CENR station (28.63°N; 17.85°W; 1208 m a.s.l.), the closest station to the Tajogaite eruption site. This station is equipped with a three-component broadband seismic sensor and has been operating continuously since 2017. Following Del Fresno et al. (2022), RSAM was calculated using dedicated software (ThomasLecocq/ssxm: RSAM/RSEM – SSAM/SSEM easy code), applying 10-minute time windows. The resulting time series was normalised by subtracting the median and dividing by the standard deviation.

#### 3.3.2 Meteorological information: Height of the Trade Wind Inversion (TWI) and wind speed/direction

Radiosonde vertical profiles are launched daily at 00:00 and 12:00 UTC from AEMET's station at Güímar (28.32°N, 16.38°W; 105 m a.s.l.). This site is a World Meteorological Organization (WMO) Global Climate Observing System (GCOS) Upper-Air Network (GUAN) station (no. 60018). This GCOS-GUAN station is located on the eastern coast of Tenerife, leeward of the prevailing trade wind flow due to the presence of a central ridge in the Güímar Valley, which reaches elevations above 2100 m a.s.l. This station has been providing upper-air profiles since 2003, measuring temperature, pressure, and humidity using Vaisala RS92 radiosondes, along with wind speed and direction obtained via a Global Positioning System (GPS) wind-finding system (Carrillo et al., 2016).

Atmospheric profiling was also conducted on La Palma (28.63°N, 17.91°W, 295 m a.s.l.) during the volcanic eruption, from 6 November 2021 to 26 December 2021 at 11 UTC, with the collaboration of AEMET and Unidad Militar de Emergencias (UME). At this temporary station, the same Vaisala RS92 radiosondes were deployed to characterise the atmosphere near the volcano. This site shares similar characteristics with the Güímar station, as it is also located on the eastern coast of La Palma, leeward of the prevailing trade winds and downstream of the island's central ridge (Cumbre Vieja), which reaches a maximum elevation of 1949 m a.s.l.

The Trade Wind Inversion height was estimated following the methodology described in Carrillo et al. (2016), based on the analysis of the temperature lapse rate, water vapour mixing ratio, and wind components on both AEMET stations (at Tenerife and La Palma).

In addition, daily wind speed and direction profiles from the AEMET numerical weather prediction (NWP) model HARMONIE-AROME (HIRLAM–ALADIN Regional/Meso-scale Operational NWP in Euromed) Version 43h2.1.1 were used. Data were extracted from the grid point closest to the eruptive centre. This model has a spatial resolution of $2.5 \times 2.5$ km$^2$, a temporal resolution of 3 h (with forecasts up to 72 h), and includes 65 vertical levels. Further details about the HARMONIE-AROME model can be found in Bengtsson et al. (2017).





### 3.3.3 Satellite derived $SO_2$ and Aerosol Layer Height (ALH)

TROPOMI (TROPOspheric Monitoring Instrument) is a sensor onboard the polar, low-Earth orbiting Copernicus Sentinel-5 Precursor (S-5P) platform, capable of providing $SO_2$ altitude layer information as a near-real-time operational product. This product is generated by the German Aerospace Center (DLR, Deutsches Zentrum für Luft- und Raumfahrt) using the Full-

Physics Inverse Learning Machine ($FP\text{-}ILM$) algorithm. The $SO_2$ layer height ($SO2LH$) product was developed by Hedelt et al. (2019) with an expected accuracy of less than 2 km for $SO_2$ vertical column densities (VCD) greater than 20 DU (Dobson Units) (Hedelt et al., 2019; Koukouli et al., 2022; Hedelt et al., 2025).

The $h_{d,\mathrm{AEMET}}$ dataset was used by Hedelt et al. (2025) to validate TROPOMI $SO2LH$ during the La Palma volcanic eruption. Altitude differences, based on median TROPOMI $SO2LH$, were found to range between 1–3 km, decreasing to

around 1 km for cloud fractions below 0.5. TROPOMI LH slightly underestimates the $h_{d,\mathrm{AEMET}}$ measurements, but the two datasets are correlated, with a Pearson correlation coefficient of $r = 0.74$ (Hedelt et al., 2025).

Another lidar sensor used in this context to provide information on the aerosol LH (ALH) is CALIOP. CALIOP is part of the CALIPSO mission providing global atmospheric profiles since June 2006 (Winker et al., 2009, 2013). This instrument orbits the Earth in a sun-synchronous orbit, acquiring lidar backscatter profiles at 532 nm and 1064 nm, including polarisation capabilities

at 532 nm (Winker et al., 2013). Level 2 version 4.2 aerosol layer information product (https://subset.larc.nasa.gov/calipso/) with a 5 km of spatial resolution was used to provide information on the aerosol layer top altitude. The CALIOP layer detection algorithm is described in detail in Vaughan et al. (2009).

### 3.3.4 Satellite-derived volcanic $SO_2$ emissions

In this study, we used the estimated volcanic $SO_2$ mass loading from the Tajogaite eruption, as provided by NASA's $MSVOLSO2L4$

multi-satellite product (Carn et al., 2016; Carn, 2022). This independent database contains all significant volcanic eruptions detected from space, offering daily estimates of volcanic $SO_2$ emissions. Since 2018, the product has relied on three spaceborne UV sensors to retrieve $SO_2$ VCD: the Ozone Monitoring Instrument (OMI) (Carn et al., 2013), the Ozone Mapping and Profiler Suite (OMPS) (Carn et al., 2015, 2016), and the TROPOspheric Monitoring Instrument (TROPOMI) (Theys et al., 2017, 2021). Infrared (IR) satellite observations have also been used to detect $SO_2$ emissions not captured by UV sensors, thereby refin-

ing the UV-based product. These IR sensors include the Atmospheric Infrared Sounder (AIRS) (Prata and Bernardo, 2007), the Moderate Resolution Imaging Spectroradiometer (MODIS) (King et al., 2003), and the Infrared Atmospheric Sounding Interferometer (IASI) (Clarisse et al., 2012), which are used for both refinement and validation purposes.

$MSVOLSO2L4$ is a multi-satellite, Level 4 version 4 volcanic $SO_2$ long-term global database product that estimates the total $SO_2$ mass (in metric tons) within a volcanic cloud by combining $SO_2$ column data with information on plume altitude.

The latter is a critical parameter, typically obtained from direct observations (e.g., pilot reports or ground-based measurements). In the absence of such observations, a default plume altitude is assumed based on the eruption style (effusive or explosive) and magnitude, inferred from the Volcanic Explosivity Index (VEI). These default altitudes can range from approximately 5 km for effusive events to 10 km for explosive eruptions.





For the initial stages of the La Palma eruption, the NASA team assumed a default injection height of 8 km, which was used in the default $SO_2$ mass loading estimates. Once a profiling network was established on La Palma, a revised product was generated for this study using the observed plume-top height ($h_{ec}$), derived on-site using the IGN video-surveillance method.

A comparative analysis between the default 8 km-based retrievals and the adjusted $SO_2$ emissions, based on ground-based plume height measurements, provides valuable insight into the impact of plume altitude assumptions on the derived $SO_2$ budgets.

## 4 Results

### 4.1 Comparison of volcanic plume height from IGN and AEMET-ACTRIS approaches

The eruptive column height measured by the IGN ($h_{ec,IGN}$) is shown in Fig. 4, along with the dispersive plume height ($h_d$) obtained from both the IGN and AEMET-ACTRIS networks ($h_{d,IGN}$ and $h_{d,AEMET}$, respectively). $h_{ec,IGN}$ and $h_{d,IGN}$ correspond to 344 measurements made by the IGN at times when a significant change in column height was observed. $h_{d,AEMET}$ (137 altitudes) were retrieved concurrently with the IGN observations using the methodology described in Sect. 3.3 for each of the five profiling stations in the network (integrated over a ±5-minute interval around the IGN observation time).

Fig. 4 shows a good agreement between the IGN and AEMET-ACTRIS datasets. The average absolute difference between $h_{d,IGN}$ and $h_{d,AEMET}$ throughout the eruptive period was 258.6 m (standard deviation of 620.4 m). This high consistency is noteworthy considering the different methodologies involved: one based on visual inspection from video-surveillance and the other one from a multi-instrumental approach including ceilometers and lidars. These differences were further analysed by station to assess their spatial dependence. The highest absolute difference was observed at ELP (941.7 m), while the lowest was recorded at RMO (139.8 m). In terms of relative differences, the mean deviation between the two datasets was 12.9%. The maximum relative difference was again found at ELP (42.7%), with the lowest at RMO (2.6%). Intermediate relative differences were observed at FUE (8.1%), TAZ (19.8%), and LPA (22.3%). The larger differences observed at ELP may be attributed to its proximity to the source (potential signal attenuation and significant impact of sudden changes in volcanic activity) or to complex topographic effects affecting the site. However, the limited number of measurements at this site prevents us from establishing any robust error metrics for comparison with other instruments in the database.

An additional insight from Fig. 4 is the relatively small difference between $h_{ec}$ and $h_d$, both measured by IGN, with an average difference of only 236.2 m. Larger differences were found in the early stages of the eruption and during the final paroxysmal event on 13 December (Del Fresno et al., 2022; Benito et al., 2023), likely reflecting more vigorous eruptive dynamics and increased column buoyancy during those phases. As a preliminary conclusion, it can be stated that both techniques provide comparable measurements and can be combined into a single data series.





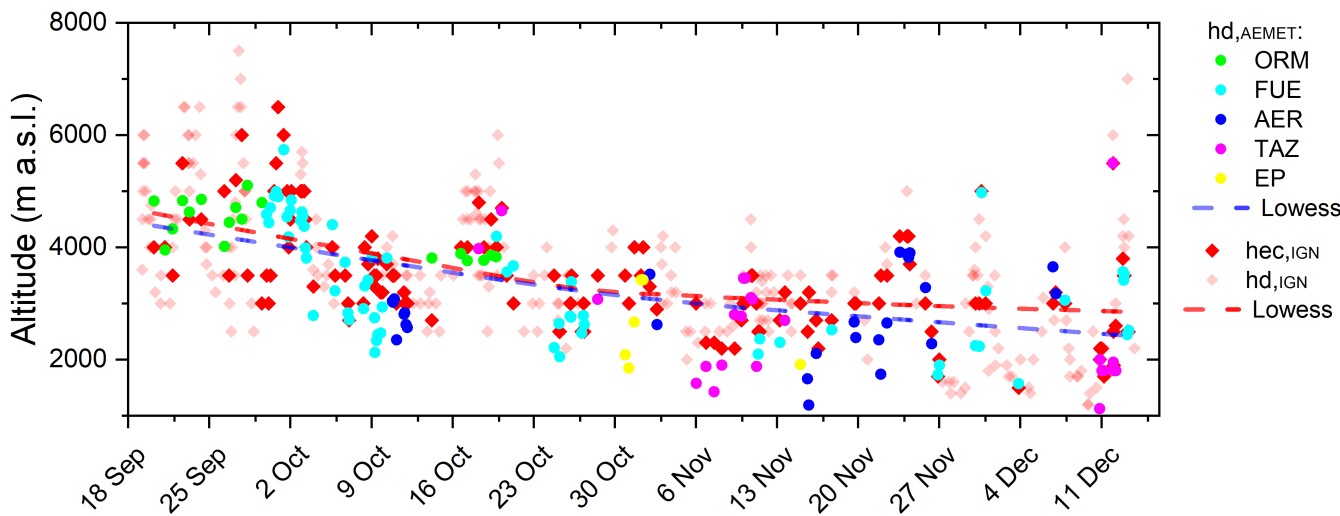

**Figure 4.** Altitude of the eruptive column in m a.s.l. measured by IGN ($h_{ec,IGN}$) and dispersion plume ($h_d$) measured by IGN ($h_{d,IGN}$) and by AEMET-ACTRIS profiling network ($h_{d,AEMET}$). Broken lines represent the corresponding lowess smoothing curves for $h_{d,IGN}$ and $h_{d,AEMET}$ datasets.

## 4.2 Description of the eruptive event in terms of tracers of the volcanic plumes ($h_d$) and modulating factors (RSAM and TWI)

The height reached by the eruptive column and its subsequent dispersion, as explained in Sect. 3.1, depends on both volcanological and meteorological parameters, as well as their interaction. According to Girault et al. (2016), key factors influencing the rise of an eruptive volcanic column include the total grain size distribution of emitted volcanic particles, the amount of gas released during magma fragmentation, and atmospheric crosswinds. Other studies (e.g., Woodhouse et al., 2016; Rossi et al., 2019) also highlight the importance of parameters such as vent radius, plume temperature at the vent, upward velocity, and the

wind entrainment coefficient.

An important modulating factor of the altitude of the eruptive column is the eruptive dynamics, which can be parameterised using proxies such as $SO_2$ emissions (Milford et al., 2023) or seismic activity (López et al., 2017; Bartolini et al., 2018). The evolution of the eruptive process during the Tajogaite eruption is shown in Fig. 5(b), represented by the RSAM signal.

Following Girault et al. (2016); Milford et al. (2023), another key factor that modulates the injection height of volcanic

emissions is the prevailing meteorological conditions. Wind speed and direction are fundamental variables that influence the direct transport and dispersion of the volcanic plume. In addition, the presence of thermal inversions has a critical effect on the vertical extent, mixing, and dispersal of volcanic emissions. As demonstrated by Milford et al. (2023), a shallower temperature







**Figure 5.** (a) Altitude of the eruptive column in m a.s.l. measured by IGN ($h_{d,IGN}$) and by AEMET-ACTRIS profiling network ($h_{d,AEMET}$). Yellow bands indicate the presence of the SAL. (b) Time series of the raw real-time seismic amplitude measurement (RSAM) measured in Cumbre Vieja by the IGN seismic network. (c) Trade Wind Inversion (TWI) height (m a.s.l.) estimated at 00:00 and 12:00 UTC from meteorological sondes launched at Güímar (Tenerife) station and at 12:00 UTC from La Palma station during the volcanic eruption (from 6 November 2021 onwards).





inversion (TWI in our case) confines the volcanic plume to lower levels of the troposphere, with significant implications for air quality and civil aviation safety.

Yellow bands in Fig. 4 indicate days affected by mineral dust conditions over La Palma, as defined by Milford et al. (2023) (26 September - 3 October; 7-8 October; 19-21 October). The presence of the Saharan Air Layer (SAL) can influence the altitude of the aerosol layer as a result of its effect on vertical stratification and atmospheric stability. According to Barreto et al. (2022), the SAL appears as a well-stratified layer capped by a temperature inversion located between $\sim 6.3$ km in summer (when the dusty air moves westwards over the subtropical North Atlantic as an elevated layer over the Marine Boundary Layer, MBL). The SAL, therefore, compresses the MBL when transported as an elevated layer.


    A comprehensive view of the eruptive process is presented in Fig. 5, which combines information on plume altitude with the evolution of the two modulating factors previously described: eruptive dynamics and meteorological conditions. This multidisciplinary approach allows for a detailed interpretation of the eruption's progression. Based on the trend in the RSAM signal (Fig. 5(b)), three distinct eruptive phases can be identified:

**Phase I** (19–27 September) is marked by alternating Strombolian and Vulcanian activity, including two strong explosive episodes on 23 and 26 September (Nogales et al., 2022), which led to the highest RSAM value recorded during the entire eruption (4.4 a.u.). This period coincides with peak values of both eruptive column height ($h_{ec}$) and dispersive plume altitude ($h_d$), reaching up to 6500 m a.s.l. and 5500 m a.s.l., respectively. Simultaneously, a downward trend in the TWI height was observed (Fig. 5(c)), with values decreasing from 2074.7 m a.s.l. to 601.4 m a.s.l. by the end of the phase. Mean TWI heights
during this period were 1576.9 m a.s.l. (12:00 UTC) and 1498.4 m a.s.l. (00:00 UTC), based on radiosonde data from Güímar. These conditions suggest a limited surface-level impact of volcanic emissions, as corroborated by surface $SO_2$ concentration and particulate matter measurements reported in Milford et al. (2023).

    **Phase II** (28 September – 2 November) is characterised by predominantly effusive activity with intermittent explosive events. The RSAM signal in this phase remains mostly between 0 and 1 a.u., with a notable increase from 24 October onwards,
reaching up to 3.1 a.u. (Nogales et al., 2022). This intensification coincides with the second-largest increase in coarse-mode aerosol fractions during the eruption (25 October – 2 November), detected via ground-based photometry (Bedoya-Velásquez et al., 2022). The TWI shows significant variability, peaking at 2780 m a.s.l. on 25 October and dropping to minimum values of 263 m a.s.l. (4 October) and 220–365 m a.s.l. (16–20 October). Average TWI heights were 1039.0 m a.s.l. and 955.9 m a.s.l. at 12:00 and 00:00 UTC, respectively. During this phase, plume heights remained relatively stable, with mean $h_d$ values of
2935.6 m a.s.l. (AEMET-ACTRIS) and 3374.4 m a.s.l. (IGN). The maximum $h_{ec}$ and $h_d$ values (5300 m a.s.l. and 4800 m a.s.l., respectively) occurred between 17–18 October.

    **Phase III** (3 November – 13 December) was dominated by low-intensity effusive activity, reflected in RSAM values generally below 0.5 a.u., with a few sharp peaks above 2 a.u. These peaks correspond to discrete events, including large ash emissions on 17 November, the opening of a new vent on 28 November, intense explosive activity with shockwaves on 1–2 December, and
the final paroxysmal phase with Vulcanian episodes on 13 December (Nogales et al., 2022). Despite the overall lower eruptive energy, this phase exhibited a stable aerosol layer and notable increases in plume heights, particularly on 12–13 December, when $h_{ec}$ reached 8500 m a.s.l. TWI values during this period were consistently high, with mean heights of 1712.5 m a.s.l.



(12:00 UTC), 1695.1 m a.s.l. (00:00 UTC) at Güímar, and 1819.3 m a.s.l. at La Palma (12:00 UTC). The intense seismic activity on 1–2 December also elevated both $h_{ec}$ and $h_d$ to approximately 5000 m a.s.l.. Observations by Bedoya-Velásquez
et al. (2022) confirmed high aerosol optical depth (AOD) coarse-mode values (0.15) during this time.

A key inflexion point between phases II and III is observed as a reduction in RSAM, also identified by Nogales et al. (2022) and Milford et al. (2023), the latter through daily $SO_2$ emissions from TROPOMI. However, Milford et al. (2023) reported that surface-level $SO_2$ concentrations did not decrease proportionally to emission rates or seismic signals. This discrepancy is attributed to the maximum TWI extension during this phase, as seen in Fig. 5(c), with mean heights ranging from 1695.1 m a.s.l.
to 1819.3 m a.s.l. This factor, combined with the low volcanic intensity, is expected to confine the volcanic plume within the lower layers of the troposphere. Consequently, this final phase of the eruption had the greatest impact on surface air quality despite exhibiting lower overall volcanic activity.

$h_{d,IGN}$ and $h_{d,AEMET}$ mean differences for the 35 days affected by the SAL are reduced to 33.4 m (1.1%), highlighting the influence of the SAL on vertical atmospheric stratification and the role of the strong temperature inversion at the top of
the SAL. This stable layer acts as a physical barrier, hindering the vertical dispersion of the volcanic plume and confining aerosols within the Planetary Boundary Layer (PBL). This situation results from the balance between the upward velocity of the volcanic plume and the strength of the inversion layer.

Another important aspect shown in Fig. 5(a) is the spatial representativity of the $h_{d,AEMET}$ measurements, which are derived from the station most affected by the volcanic plume depending on prevailing wind direction. The spatial distribution
of these data reveals a significant bias toward the western side of La Palma Island, as expected, given the predominant trade wind regime (more than 63.5% of the data in the $h_{d,AEMET}$ time series were recorded at the Tazacorte and Fuencaliente stations). Specifically, the number of plume height measurements per station is as follows: 18 from ORM (mainly restricted to Phase I), 24 from TAZ, 63 from FUE, 6 from ELP, and 26 from LPA. This distribution highlights the role of wind direction in plume transport and vertical dispersion, further evidenced in Fig. 6, which displays the wind rose diagram for the study period.

## 4.3 Comparison between $h_{d,AEMET}$ with CALIOP aerosol altitude ($ALH_{CALIOP}$)

A total of 12 CALIOP overpasses during the volcanic eruption were used in this study in the evaluation of the CALIOP level 2 layer information product. The different altitudes ($ALH_{CALIOP}$) corresponding to the highest aerosol layer detected by CALIOP are shown in Fig. 7 together with the $h_{d,AEMET}$ series. A mean difference of 615.0 m was retrieved with a clear underestimation of the CALIOP altitudes with respect to the ground-based measurements. These overpasses were done at an
average distance of 61.7 km from the volcanic vent, and the AEMET-ACTRIS versus CALIOP differences do not seem to be dependent on the overpass distance. The maximum difference was observed on 6 October (a discrepancy of 3065.5 m, with an overpass distance of 28 km). The aerosol layer altitude difference on this particular day is attributed to the presence of a thin aerosol layer located between approximately 2 and 4 km, as observed by the CHM15k ceilometer at FUE, with the PBL situated below (not shown for the sake of brevity). It appears that CALIOP was unable to detect weak backscatter signals,
presumably because they were below its detection threshold. Excluding this day, the mean difference is reduced to 392.2 m. These differences are consistent with previous studies using ground-based lidar measurements (Perrone et al., 2011; Kim et al.,





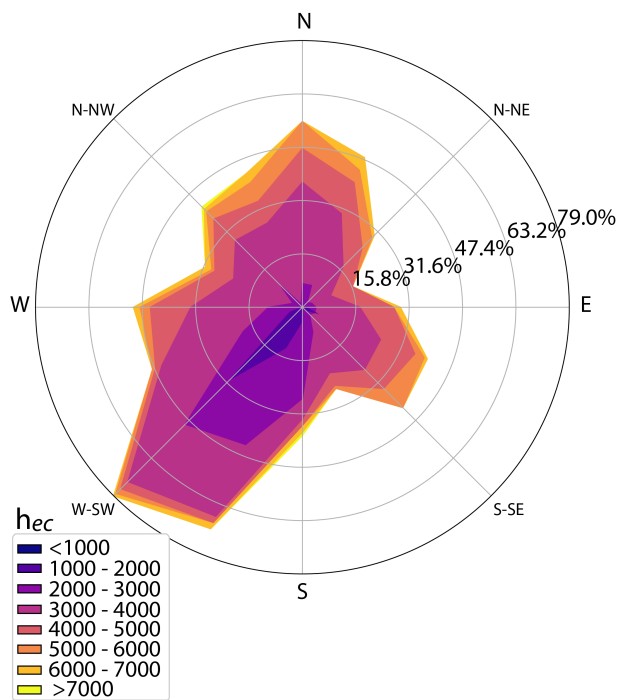

**Figure 6.** Wind rose diagram for the volcanic eruptive column $h_{ec}$ (in m a.s.l.) and HARMONIE-AROME wind vectors at the pixel of the volcanic edifice in Cumbre Vieja. Contour lines are represented with the frequency of occurrence in %.

2008), as well as with results obtained using other techniques (Hedelt et al., 2019; Tournigand et al., 2020; Nanda et al., 2020; Chen et al., 2020).

## 4.4 Impact of the altitude of the eruptive plume on satellite-based $SO_2$ emission estimation

As already stated in the first part of this paper, the height of the volcanic plume is an important factor to take into account when it comes to estimating the $SO_2$ emission loadings using satellite observations. Fig. 8 shows the evolution of the eruptive event in terms of daily $SO_2$ emission (in kt) assuming a reference $h_{ec}$ of 8 km for effusive volcanoes such as Tajojaite. In this figure, $SO_2$ estimations using $h_{ec,IGN}$ observations are also included (standard $MSVOLSO2L4$ NASA product). The change in the eruptive activity is apparent from this figure, showing a strong decrease in the measured volcanic $SO_2$ emissions during the

first days of November. Average $SO_2$ during the first phase was 23.8 kt (42.9 kt in the case of the corrected series using the $h_{ec,IGN}$), in the second phase was 17.3 (44.3) kt, while this average in the case of the last phase is reduced to 4.7 (10.1) kt. This decrease is attributed to the change in the eruptive activity observed from the RSAM series in Sect. 4.2.

Regarding the impact of using a real $h_{ec}$ instead of an a priori value (8 km in the case of La Palma eruption), we observe significant subestimation of the $SO_2$ emission mass estimated from the satellite. An average $SO_2$ mass emission difference of




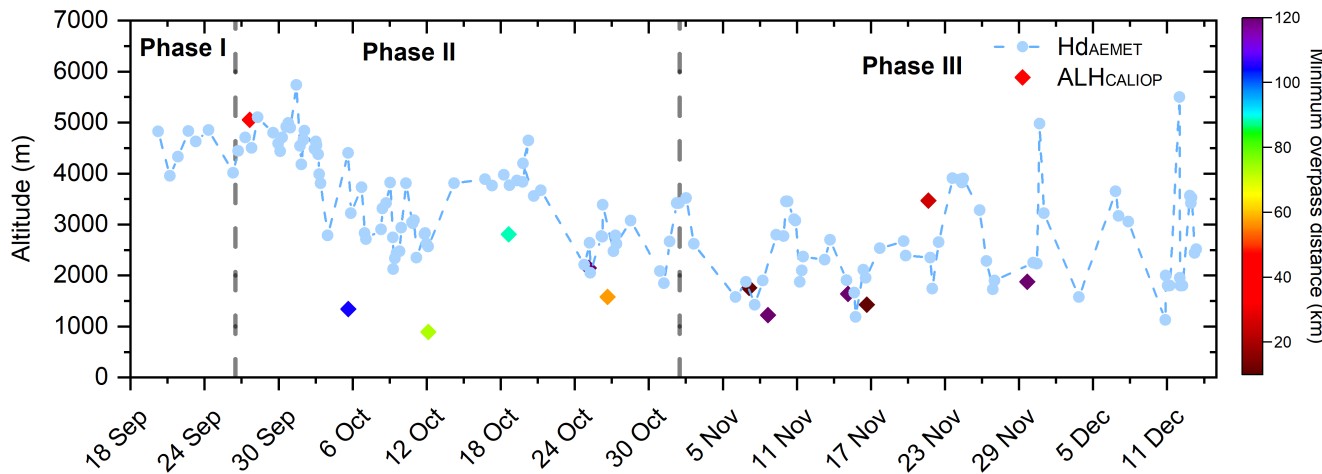

**Figure 7.** Altitude of the dispersion plume $h_{d,AEMET}$ (in m) and aerosol layer height determined from CALIOP ($ALH_{CALIOP}$) measured during the entire volcanic eruption, with the distance of the overpass from the volcanic vent (in km) in the colorbar.

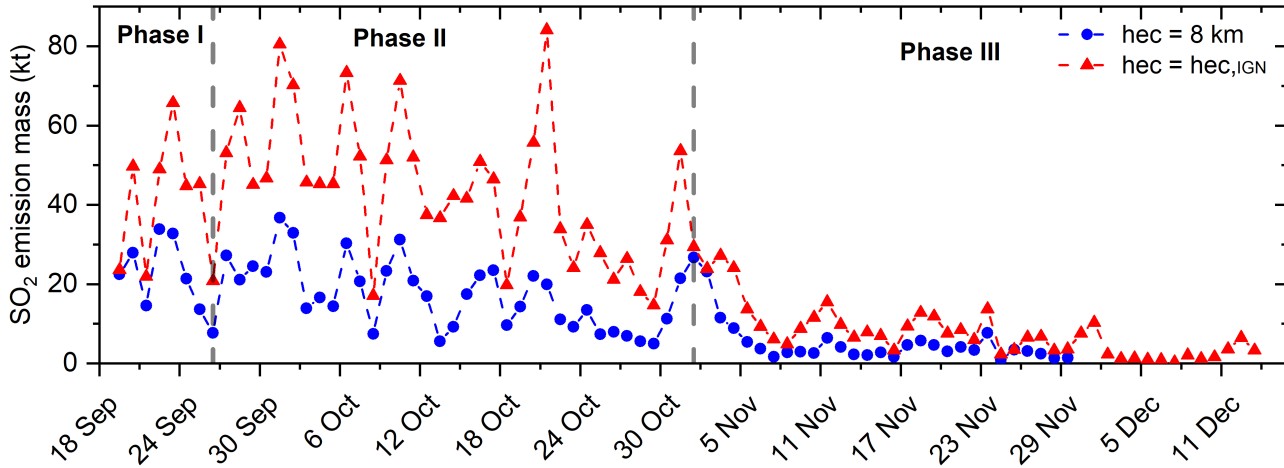

**Figure 8.** NASA $SO_2$ emission mass (in kilotons, $kt$) calculated as the average of OMI, OMPS and TROPOMI UV backscatter radiances considering a standard columnar injection height of 8 km (orange) and the real $h_{ec}$ measured by the IGN (in light red). Dotted vertical lines represent the three eruptive phases derived from RSAM data series.

17.3 kt has been found, with maximum values up to 64.2 kt (found on October 22). Average relative differences of 56.2% have been found in the whole eruptive process, with maximum values found on October 13 (84.7%).





## 5 Summary and Conclusions

This work provides a comprehensive description of the eruptive process that took place on La Palma Island (19 September – December 2021) from a different scientific perspective, offering complementary information and novel insights to those of previous studies. Regarded as the most significant eruption in Europe over the past 75 years in terms of $SO_2$ emissions and societal impact, the Tajogaite eruption provided a unique natural laboratory to study explosive volcanic activity from multiple scientific perspectives. In our case, this was made possible through an unprecedented collaborative effort among public institutions, research groups, and private entities, coordinated by IGN and AEMET, the latter within the framework of ACTRIS and ACTRIS-Spain (ACTRIS 2021).

The characterisation of the eruption presented in this study focused on key plume tracers, including the height of the eruptive column ($h_{ec}$) and the dispersive volcanic plume ($h_d$), as well as modulating factors such as seismic activity and meteorological conditions. Based on the temporal evolution of the RSAM signal, three distinct eruptive phases were identified. These phases reflect changes in the eruptive dynamics, plume altitude, and associated seismic and atmospheric conditions throughout the event. In this regard, our results reveal a complex eruptive process characterised by a range of eruptive styles, from explosive Strombolian phases to predominantly effusive activity.

Two independent and complementary observational techniques were employed to monitor the vertical evolution of the plume: the video-surveillance system operated by IGN and the profiling network deployed by AEMET-ACTRIS, which includes ceilometers and lidars positioned strategically around the volcanic vent. This key information was incorporated into the PEVOLCA reports (in the case of both IGN and AEMET-ACTRIS datasets) during the whole volcanic crisis. The VONA alerts and the regular reports submitted to the Toulouse VAAC were informed by the IGN dataset.

These two techniques present intrinsic limitations. The first method, based on visual observations from the IGN camera network, is generally not applicable during nighttime hours. Nevertheless, during the most intense eruptive phases, nighttime observations were feasible due to the high luminosity of the eruptive column. Additionally, under conditions of dense cloud cover or fog, the top of the plume was often obscured, making accurate estimations difficult. The second method, based on vertical profiling using ceilometers and lidars (AEMET-ACTRIS network), only measures the atmosphere directly above each station. Consequently, plume detection is limited to instances when the volcanic layer is present in the station's vertical column. Moreover, the AEMET-ACTRIS dataset integrates data from multiple instruments and locations, which are differently influenced by the prevailing East-to-West trade wind pattern—resulting in a spatial sampling pattern, with over 63.5% of data originating from Tazacorte and Fuencaliente on the western flank.

The relatively low overall discrepancies found in this study (258.6 m) confirm the coherence between the two retrieval techniques. This agreement is noteworthy given the different methodologies involved, and it demonstrates that both techniques can yield comparable volcanic plume height measurements, which may be combined into a single data series. This approach may be of use in future volcanic crises and could support operational surveillance during such events.

The $h_{d,AEMET}$ dataset was also used to evaluate satellite-derived plume altitudes near the source. Comparisons with the CALIOP aerosol layer height ($ALH_{CALIOP}$) showed a mean underestimation of 615.0 m. This value can be reduced to



392.2 m if one specific day, characterised by the presence of a thin aerosol layer, is excluded from the comparison. In addition to their inherent operational use in volcanic monitoring, we have demonstrated in this work that the $h_d$ and $h_{ec}$ products also provide an opportunity to evaluate different satellite-derived datasets, which is crucial for improving their applicability in future volcanic crises.

The impact of assuming a fixed plume height in the estimation of satellite-based $SO_2$ emissions was also assessed. Two versions of NASA's $MSVOLSO2L4$ product were analysed: (1) an initial estimation assuming a fixed plume height of 8 km above the vent, and (2) a final version using actual $h_{ec}$ values provided by IGN. Our results indicate that assuming a fixed plume height leads to significant underestimations of $SO_2$ mass loadings, with average discrepancies of 56.2% over the eruption and peak deviations reaching 84.7%. These findings underscore the critical importance of accurate plume height estimation when

quantifying volcanic emissions using satellite data.

*Data availability.* Data from MPLNET used in the present study can be obtained from https://mplnet.gsfc.nasa.gov/download_tool/ (accessed on Feb 21, 2025). The vertical meteorological soundings can be downloaded from http://weather.uwyo.edu/upperair/sounding.html (accessed on March 14, 2025). MSVOLSO2L4 is accessible at https://disc.gsfc.nasa.gov/datasets/MSVOLSO2L4_4/summary (accessed on April 15, 2025). Level 2 version 4.2 CALIOP aerosol layer information product is available at https://subset.larc.nasa.gov/calipso/(accessed

on March 14, 2025). All the information about the height of the volcanic plume will be available upon request to the contributing authors.

*Author contributions.* AB and OG provided the volcanic plume height from the AEMET-ACTRIS network. FQ and JP-P provided the height from the IGN network. DG did the data analysis concerning CALIOP and wrote and revised the paper. AB-V provided the volcanic height from CL51 and CL61 profilers; MS and CC-J provided the altitudes from MPL-4B; MI and VR provided the height of the volcanic plume from ARCADE; and NK and SC provided the $SO_2$ mass emission estimates from satellites. The remaining authors, listed in alphabetical

order, actively contributed scientifically to the development of the paper, either through fieldwork during the eruption or by supporting the retrieval of the volcanic plume height series throughout the three-month eruptive period. Furthermore, EW provided calibrated data of the MPL lidar; AF, SM, SC and NK also contributed to the writing and the discussion on the data. PH and DL provided insight on TROPOMI data. MH was in charge of vertical soundings at La Palma during the eruption. All authors were involved in helpful discussions and contributed to the manuscript.

*Competing interests.* The authors have the following competing interests: At least one of the (co-)authors is a member of the editorial board of Atmospheric Measurement Techniques.

*Acknowledgements.* This work is part of the activities of the WMO-Measurement Lead Centre for aerosols and water vapour remote sensing instruments (MLC). Scientific activities in this paper were done under the ACTRIS grant (agreement no. 871115), ACTRIS ERIC Spain Grant



RED2024-153891-E funded by MICIU/AEI/10.13039/501100011033 and other grants supported by the Ministerio de Ciencia e Innovación
(MICINN) (PID2023-151666NB-I00 and PID2021-127588OB-I00). Financial support of the Department of Education, Junta de Castilla y
Leon, and FEDER Funds is gratefully acknowledged (Reference: CLU-2023-1-05). M. Sicard is supported by the Horizon Europe European
Research Council (project REALISTIC, grant no. 101086690). This research has also been supported by COST (European Cooperation in
Science and Technology) under the HARMONIA (International network for harmonization of atmospheric aerosol retrievals from ground-
based photometers) Action CA21119.

We gratefully acknowledge the data provided by the MPLNet network. The MPLNET project is funded by the NASA Radiation Sciences
Program and the Earth Observing System.

The authors gratefully acknowledge the extraordinary effort carried out by the AEMET staff (both in La Palma and in support of the
activities in La Palma) during the volcanic eruption, from the Izaña Atmospheric Research Center and the Delegation of AEMET in the
Canary Islands. We also gratefully acknowledge the dedication of Dr. David Suárez and all the information provided by the PEVOLCA
Scientific Committee, in adittion to the support received from the insular and local governments (Cabildo Insular de La Palma and the
Ayuntamientos de Tazacorte, Los Llanos de Aridane and El Paso).

Finally, the authors would like to thank Dr. Emilio Cuevas for his support and leadership in this project, making the necessary institutional
collaboration possible to carry out the measurements presented in this paper.



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
