# Peer review of "Volcanic plume height during the 2021 Tajogaite eruption (La Palma) from two complementary monitoring methods. Implications for satellite-based products"

_EGUsphere, 2025_

## Author Comment (AC1)

**GENERAL COMMENTS**

The study by Barreto et al. compares estimates of plume height during the Tajogaite eruption in 2021 derived from two independent approaches: one based on video-surveillance cameras and the other on ground-based remote-sensing profiling instruments. The temporal evolution of plume height is further examined in relation to ancillary measurements. The implications of plume height overestimation for satellite-based $SO_2$ retrieval algorithms are considered, and plume altitude estimations from the surface are compared with CALIOP observations.

The manuscript presents a substantial amount of material, and the overall analyses appear solid and convincing. However, the paper lacks a clear articulation of the specific research questions, which makes it somewhat difficult to follow and gives the impression that the central thread of the study is not fully established. In addition, a more detailed discussion of the uncertainties associated with each technique would strengthen the comparison of results. Overall, I would recommend publication once these issues have been adequately addressed.

**Answer from the authors to the general comments:** The authors would like to express their gratitude for the constructive and insightful comments provided by this referee. They have undoubtedly contributed to improving the quality of the manuscript and significantly enhancing its overall organization.

The specific and technical comments are addressed in detail in the following sections.

**SPECIFIC COMMENTS**

**1. Research questions – Please state the main research questions explicitly in the Introduction, and organise the manuscript accordingly. In addition, highlight the novel aspects of the study in relation to the abundant literature on the same eruption already cited in the bibliography.**

The authors agree with this referee's comment and would like to thank him/her for the opportunity to improve such an important part of the manuscript as the scientific introduction. The original text contained a considerable amount of information on volcanic aerosols which, although being the main focus of this study, represents knowledge already well established in the scientific literature. Therefore, the amount of information on volcanic aerosols has been reduced in the second paragraph.

The introduction now has been refocused on the general challenges involved in measuring volcanic aerosol. These challenges are common to both stratospheric injections and tropospheric ones (arising from smaller eruptions or hydrothermal

degassing processes), although the latter have been less frequently investigated in the literature, as their effects are often considered "minor."

In line with this argument, the deployment carried out during the La Palma eruption is presented as a relevant scientific exercise to investigate this type of volcanic activity through two key parameters: the monitoring of the volcanic plume height and its impact on the quantification of volcanic emissions from satellite observations.

**2. Algorithm consistency – If I understand correctly, different algorithms were used to estimate plume height from profiling instruments. Why was a single, uniform algorithm not applied, which would have enabled a more consistent comparison?**

The authors agree with the referee's assessment that, in a standard intercomparison of instruments, it would be advisable for all of them to operate under similar conditions and, for the sake of consistency, to use the same algorithms for estimating the volcanic plume height.

However, as outlined in the scientific objectives of this paper, the present study uses data provided by the different institutions that deployed their instruments during the La Palma eruption. The goal is to demonstrate the potential of combining and complementing various measurement techniques and methods to enhance the available information on the same physical phenomenon, namely the volcanic plume height. Most of the individual data series have already been published and were obtained using the algorithm that, according to each institution, is considered the most appropriate for the specific characteristics of their profiler. We have therefore combined individual datasets that provided quasi near-real-time measurements during the eruption, which contributed to validating the datasets delivered by the IGN, as the competent authority. A subsequent comparison has been performed to assess whether these two sources of information (video-surveillance cameras versus ground-based profiling) can be considered comparable and applicable to future volcanic crises.

From the authors' point of view, the multi-instrumental (and inter-method) approach presented in this article holds scientific interest in the context of unpredictable events, such as a volcanic eruption, where diverse and heterogeneous observations can be combined and prove useful. As demonstrated in the specific case of the La Palma eruption, such an approach can support civil protection, aviation safety, and air quality management efforts.

Nonetheless, studies published in the literature have compared the methodologies generally applied in this study and concluded that they can be regarded as equivalent and reliable, as stated in the manuscript.

**3. Uncertainty – A thorough discussion of the uncertainties associated with plume altitude retrievals from the different methods is essential before drawing comparisons. Without this, the statement that the results are "highly consistent" (lines 393–394) is difficult to justify.**

The authors agree on the need to incorporate additional statistical indicators to support the statement of "high consistency" between the two datasets. In fact, this

issue was also raised by Referee 2. The following indicators have been included (in the new Table 1) in accordance with the new text added to the manuscript:

[Line 402] In light of the comparison results shown in Table 1, it can be observed that the methodology used to derive the volcanic plume height from lidar data does not appear to play a dominant role in the comparison outcomes, with the best statistics obtained for the FUE and TAZ cases (Gradient Method in both cases).

Table 1: Main statistical skill scores (in m) for the comparison between $h_{d,IGN} - h_{d,AEMET}$ differences, including also the multi-instrument and inter-method comparison between $h_{d,IGN}$ and the height of the volcanic plume measured by AEMET at the five different stations. The methodology for retrieving the altitude of the volcanic plume is also included. GM stands for the Gradient Method, and WCT for the Wavelet Covariance Transfer method.

|  | $h_{d,IGN} - h_{d,AEMET}$ | $h_{d,IGN} - h_{d,ORM}$ | $h_{d,IGN} - h_{d,FUE}$ | $h_{d,IGN} - h_{d,AER}$ | $h_{d,IGN} - h_{d,TAZ}$ | $h_{d,IGN} - h_{d,EP}$ |
|---|---|---|---|---|---|---|
| Methodology | Multi-approach | Qualitative | GM | WCT | GM | WCT |
| Mean Difference | 258.6 | -139.8 | 203.6 | 430.4 | 344.8 | 941.7 |
| Standard Deviation | 620.4 | 866.5 | 531.7 | 540.7 | 487.1 | 468.5 |
| RMSE | 672.1 | 877.7 | 569.4 | 691.1 | 596.8 | 1051.8 |
| Pearson coefficient (r) | 0.81 | 0.18 | 0.86 | 0.64 | 0.89 | 0.72 |
| Slope | 0.90 | 0.10 | 0.86 | 0.99 | 0.99 | 1.16 |
| Intercept | 103.4 | 3918.8 | 307.1 | -426.4 | -302.9 | -1520.6 |

Furthermore, the distinction in this table regarding the type of methodology used for each AEMET station addresses the technical issue raised by this referee concerning Section 4.1, where it is noted that the discussion should be differentiated based on the methodology type, rather than focusing on the research group or measurement station.

**Moreover, it is unclear whether plume height is always defined as the altitude of the plume top, or alternatively as the altitude corresponding to a maximum in the signal. Please clarify.**

In the case of the profiler network, the methodology applied, such as the Gradient Method, refers to the estimation of the volcanic plume top. These methods detect strong changes in the vertical gradient of the signal in order to discern the transition point between the layer and the clean atmosphere. In the case of the WCT method, the principle is also to identify abrupt changes in the vertical profile, but using a wavelet transform, which highlights structures at different scales. It also detects the top of the layer at the point of maximum covariance, where there is a strong change in the signal. The video-surveillance network method relies on visual observations using calibrated images employed to detect the top of the eruptive column and the height of the dispersing column. In this way, all the techniques used in the current study are performing the estimation of the volcanic plume top.

**4. Modulating factors – The discussion of the relationships between plume height and ancillary variables is predominantly qualitative. Are there any models, even empirical ones, that could provide more quantitative estimates of plume altitude as a function of the parameters considered?**

Different models, exhibiting varying levels of complexity, have been developed in the scientific literature to derive eruption parameters, as the one plume height in our study. These models require accurate information on the mass eruption rate, vertical profiles of ash injection, grain-size distribution, plume height, ash-column dynamics, and meteorological conditions, among other factors (Plu et al., 2021).

According to Plu et al. (2021), the volcanic source term — defined as the mass of ash injected into the atmosphere as a function of height and time — is a key variable that is prone to large uncertainties. The use of a resolved source term, which explicitly simulates the thermodynamic and buoyancy processes within the plume, provides a more realistic representation of the horizontal dispersion of volcanic ash. This is the case for the 1D models Plumeria (Mastin, 2007) and FPlume (Folch et al., 2016), which include microphysical aerosol processes and aggregation. These models are intended to be user-friendly and easy to run, although a large amount of input information is required.

The input parameters of this type of models typically include:

- Atmospheric properties: air temperature at the vent, relative humidity, tropospheric lapse rate, tropopause elevation, thickness of the isothermal layer, stratospheric lapse rate, and wind speed;
- Vent properties: vent elevation, diameter, exit velocity, and mass fraction of added water;
- Magma properties: magma temperature, gas mass fraction, specific heat, and density.

In the case of FPlume, additional details are required, such as terminal settling velocity, aggregate characteristics, and particle size distribution, among others.

Taking into account this information, the authors agree with the referee that the use of a volcanic plume model to provide more quantitative estimates and to explore the role of modulating factors in greater depth would represent a clear added value to this study. However, the lack of some necessary input information for these models prevented us from performing this specific analysis in the present work. This represents an ongoing line of research that will require input from another research team working on the island during the eruption, who will be able to provide the required model input data, especially regarding vent and magma properties.

Another model used in the literature to estimate the plume heigh is Plumetraj, a pixel-based trajectory analysis of the $SO_2$ cloud (Pardini et al., 2017). PlumeTraj takes the $SO_2$ imagery from TROPOMI and tracks the plume back to the volcano to calculate the sub-daily $SO_2$ emission as a function of time and altitude. Esse et al. (2025) has recently published the altitude of the $SO_2$ plume in the Tajogaite eruption by using PlumeTraj, showing injection altitudes matching well with those measured on the ground (from PEVOLCA and VONA, i.e., from IGN measurements). However, a detailed statistics analysis on quantifying these differences is lacking in this article, which is more focussed on $SO_2$ emission rates. The information regarding the comparison in terms of plume altitude will be included in the corrected manuscript on line 502 (in addition to references to other ground-based observations required by the Referee 2):

"Consistent daily emissions for the Tajogaite volcano were calculated by Esse et al. (2025) by means of a novel forward trajectory approach using PlumeTraj, a pixel-based trajectory analysis of an $SO_2$ cloud (Pardini et al., 2017). Ground-based miniDOAS observations of $SO_2$ emissions were also carried out during the Tajogaite eruption. This is exemplified by the studies of Albertos et al. (2022) and Rodríguez et al. (2023), which reported $SO_2$ emissions ranging from 670 to 17 tons per day and observed a clear decreasing trend in $SO_2$ during the post-eruptive phase."

The information regarding the PlumeTraj $SO_2$ emissions will be discussed below, in the Technical Question number 6 of this referee.

References:

Mastin, L. G.: A user-friendly one-dimensional model for wet volcanic plumes, Geochem. Geophy. Geosy., 8, Q03014, https://doi.org/10.1029/2006GC001455, 2007.

Folch, A., Costa, A., and Macedonio, G.: FPLUME-1.0: An integral volcanic plume model accounting for ash aggregation, Geosci. Model Dev., 9, 431–450, https://doi.org/10.5194/gmd-9-431-2016, 2016.

Plu, M., Bigeard, G., Sič, B., Emili, E., Bugliaro, L., El Amraoui, L., Guth, J., Josse, B., Mona, L., and Piontek, D.: Modelling the volcanic ash plume from Eyjafjallajökull eruption (May 2010) over Europe: evaluation of the benefit of source term improvements and of the assimilation of aerosol measurements, Nat. Hazards Earth Syst. Sci., 21, 3731–3747, https://doi.org/10.5194/nhess-21-3731-2021, 2021.

Esse, B., Burton, M., Hayer, C., La Spina, G., Cofrades, A. P., Asensio-Ramos, M., Barrancos, J., and Pérez, N.: Forecasting the evolution of the 2021 Tajogaite eruption, La Palma, with TROPOMI/PlumeTraj-derived $SO_2$ emission rates, Bulletin of Volcanology, 87, 20, https://doi.org/10.1007/s00445-025-01803-6, 2025.

Pardini, F., Burton, M., de' Michieli Vitturi, M., Corradini, S., Salerno, G., Merucci, L., and Di Grazia, G.: Retrieval and intercomparison of volcanic $SO_2$ injection height and eruption time from satellite maps and ground-based observations, Journal of Volcanology and Geothermal Research, 331, 79–91, https://doi.org/https://doi.org/10.1016/j.jvolgeores.2016.12.008, 2017.

**5. CALIOP – An average overpass distance of 61.7 km appears rather large for direct comparison with CALIOP. How far is the plume expected to be horizontally transported under the observed conditions?**

Following Carn et al. (2016), CALIOP detects aerosol particles and not $SO_2$ gas. However, it is often assumed that aerosols and $SO_2$ will be approximately collocated (horizontally and vertically). Clarisse et al. (2014) found good agreement between IASI $SO_2$ altitude retrievals and CALIOP aerosol layer altitudes for the 2011 Nabro eruption cloud. In this study, Clarisse et al. (2014) used satellite measurements in a 0.5 degrees radius along the CALIPSO overpass, which means distances around 55 km.

There is no consensus in the literature regarding the maximum distance beyond which it becomes meaningless to compare ground-based and satellite measurements. In the case of volcanic eruptions, it is known that plume dispersion processes and the conversion of $SO_2$ to sulfate can play an important role when selecting the CALIOP overpass distance. This factor is usually considered as a source of uncertainty in such comparisons; however, the limited availability of data

must also be taken into account as an alternative factor to be minimized, in order to ensure a sufficient number of data points for a meaningful comparison. Hence, the selection of CALIOP overpasses should aim to achieve an optimal trade-off between obtaining a statistically meaningful number of coincidences and preserving a sufficiently small spatial separation, so that the compared datasets remain physically representative of the same atmospheric scene, as it is done in the present study.

Regarding the specific question raised by this Referee about how far is expected to be horizontally transported the plume, we present in the following figures SO$_2$ concentrations measured by the TROPOMI sensor (extracted from VolcPlume Portal, https://volcplume.aeris-data.fr) for each of the overpasses selected for the present study. It can be seen that the distribution of the volcanic plume extends approximately from 150 km to several thousand kilometers. Although at large distances from the vent the presence of SO$_2$ cannot be directly attributed to sulfate aerosol due to chemical conversion, sedimentation, or other atmospheric mixing processes, it appears plausible to make such a comparison at shorter distances, as considered in the present work.

[Figure]

[Figure]

References:

Carn, S., Clarisse, L., and Prata, A.: Multi-decadal satellite measurements of global volcanic degassing, Journal of Volcanology and Geothermal Research, 311, 99–134, https://doi.org/https://doi.org/10.1016/j.jvolgeores.2016.01.002, 2016.

Clarisse, L., Coheur, P.-F., Theys, N., Hurtmans, D., and Clerbaux, C.: The 2011 Nabro eruption, a SO2 plume height analysis using IASI measurements, Atmos. Chem. Phys., 14, 3095–3111, https://doi.org/10.5194/acp-14-3095-2014, 2014.

**Furthermore, what are the results of the CALIOP aerosol typing? To which aerosol class is the layer attributed?**

It is important to mention that, in the CALIPSO version 4 automated aerosol classification, as defined by Kim et al. (2018), volcanic aerosols are only represented as stratospheric aerosols. This is the reason for not including this typing in the manuscript. In the case of CALIOP typing algorithm, following Kim et al. (2018), volcanic sulfate within the troposphere will be assigned a tropospheric aerosol type, usually elevated smoke or clean continental if weakly scattering, or can be misclassified as dust and polluted dust, due to elevated depolarization.

In the figures below, the Referee can found the different aerosol classification performed by CALIOP of the different overpasses used in this study, mainly dust or smoke.

[Figure]

[Figure]

References:

Kim, M.-H., Omar, A. H., Tackett, J. L., Vaughan, M. A., Winker, D. M., Trepte, C. R., Hu, Y., Liu, Z., Poole, L. R., Pitts, M. C., Kar, J., and Magill, B. E.: The CALIPSO version 4 automated aerosol classification and lidar ratio selection algorithm, Atmos. Meas. Tech., 11, 6107–6135, https://doi.org/10.5194/amt-11-6107-2018, 2018.

**6. Emissions – Is there any means of determining whether the new estimates of SO$_2$ emissions based on measured plume altitude provide an improvement over the default values?**

Since plume altitude is one of the main sources of uncertainty in satellite retrievals of volcanic SO$_2$, any information on actual (i.e., measured) plume altitude is always expected to improve the retrievals relative to the default assumption (a fixed plume altitude). This is particularly important for lower tropospheric SO$_2$ plumes (< 5 km altitude), as observed during the La Palma eruption, since satellite sensitivity to SO$_2$ declines rapidly towards the surface.

Following Theys et al. (2013), and already stated in the paper, one of the largest errors on the SO$_2$ columns is due to a poor a priori knowledge of the height of the SO$_2$ plumes. Following Carns et al. (2016), SO$_2$ retrieval algorithms require specification of SO$_2$ altitude due to the temperature- and pressure-dependence of SO$_2$ absorption and the air mass factor (the ratio of slant to vertical column density). It is assumed that, if the true altitude of the plume is lower than the assumed value, which is the current case of this paper (standard altitude of the satellite product set at 8 km), then the SO$_2$ column amount (and therefore the emission fluxes) will be underestimated (Taylor et al., 2018). This is the result observed in Figure 8 and Sect. 4.4. Having said that, the attribution in the paper that the new product, created using the real (measured) plume altitude, leads to an improvement in the satellite retrievals is reasonable. However, as the referee notes, new estimates of SO$_2$ emissions would be helpful to substantiate this conclusion.

We found relevant results to perform this comparison in the daily mean SO$_2$ emissions provided by Esse et al. (2025). These results are based on PlumeTraj, which uses SO$_2$ imagery from TROPOMI and traces the plume back to the volcano to calculate sub-daily SO$_2$ emissions as a function of time and altitude. Although this is not a ground-based validation, it has been published as an improvement over standard techniques for estimating plume altitude. The comparison is shown in the following figure:

[Figure]

While relative differences between $h_{ec}$ and $h_{8km}$ of 56.2% were found in the whole eruptive process, these differences have been reduced to 21.6% when $h_{ec}$ and $h_{PlumeTraj}$ were compared. These results confirm the better performance of the two new products (real altitudes and altitude retrieved by means of PlumeTraj).

The following information has been included in the corrected manuscript (line 506):

"Comparable daily $SO_2$ emission values to those retrieved using $h_{ec,IGN}$ were reported by Esse et al. (2025) for the same eruption, based on the PlumeTraj analysis, with lower mean relative differences of 21.6%."

It is indeed advisable to perform a direct validation using $SO_2$ emission data obtained from ground-based measurements. Although numerous attempts were made during the volcanic eruption to carry out such measurements, only a few publications have been released on this topic. Among them, Taquet et al. (2025) stand out, reporting direct-sun measurements using low- (EM27/SUN) and high- (IFS-125HR) spectral resolution Fourier Transform InfraRed (FTIR) spectrometers located up to ~140 km away from the volcano. These are, however, column-integrated measurements that would require complex wind and plume geometry modeling to be converted into emission fluxes.

Another interesting contribution comes from the study by Birnbaum et al. (2023), who provided apparent volume flux data (m³/s) acquired during the eruption. These data would need to be converted into emission fluxes, a process that would involve complex calculations beyond the scope of the present work.

Direct measurements were also performed using the DOAS (Hayer et al., 2022) or mini-DOAS (Albertos et al., 2022; Rodríguez et al., 2023) techniques by several scientific groups, but no results from such observations have yet been published, highlighting the challenges of extracting accurate information from this type of measurement.

Other proxies could be used to obtain rough estimates of $SO_2$ emission fluxes, such as lava flow area or $CO_2$ flux data, which should correlate to some extent with $SO_2$ emissions.

However, given the complexity of these analysis, the lack of published dataset, and the expected high uncertainty in the potential comparison with proxies, the authors consider that existent ground-based observations would not be an appropriate approach to test the consistency of the satellite-derived emission fluxes presented in this work.

References:

Albertos, V. T., Recio, G., Alonso, M., Amonte, C., Rodríguez, F., Rodríguez, C., Pitti, L., Leal, V., Cervigón, G., González, J., Przeor, M., Santana-León, J. M., Barrancos, J., Hernández, P. A., Padilla, G. D., Melián, G. V., Padrón, E., Asensio-Ramos, M., and Pérez, N. M.: Sulphur dioxide ($SO_2$) emissions by means of miniDOAS measurements during the 2021 eruption of Cumbre Vieja volcano, La Palma, Canary Islands, EGU General Assembly 2022, Vienna, Austria, 23–27 May 2022, EGU22-5603, https://doi.org/10.5194/egusphere-egu22-5603, 2022.

Birnbaum, J., "Temporal variability of explosive activity at Tajogaite volcano, Cumbre Vieja (Canary Islands), 2021 eruption from ground-based infrared photography and videography", Frontiers in Earth Science, vol. 11, Art. no. 1193436, 2023. doi:10.3389/feart.2023.1193436.

Hayer, C., Barrancos, J., Burton, M., Rodríguez, F., Esse, B., Hernández, P., Melián, G., Padrón, E., Asensio-Ramos, M., and Pérez, N.: From up above to down below: Comparison of satellite- and ground-based observations of $SO_2$ emissions from the 2021 eruption of Cumbre Vieja, La Palma, EGU General Assembly 2022, Vienna, Austria, 23–27 May 2022, EGU22-12201, https://doi.org/10.5194/egusphere-egu22-12201, 2022.

Rodríguez, O., Barrancos, J., Cutillas, J., Ortega, V., Hernández, P. A., Cabrera, I., and Pérez, N. M.: $SO_2$ emissions during the post-eruptive phase of the Tajogaite eruption (La Palma, Canary Islands) by means of ground-based miniDOAS measurements in transverse mode using a car and UAV, EGU General Assembly 2023, Vienna, Austria, 23–28 Apr 2023, EGU23-3620, https://doi.org/10.5194/egusphere-egu23-3620, 2023.

Taquet, N., Boulesteix, T., García, O., Campion, R., Stremme, W., Rodríguez, S., López-Darias, J., Marrero, C., González-García, D., Klügel, A., Hase, F., García, M. I., Ramos, R., Rivas-Soriano, P., Léon-Luis, S., Carreño, V., Alcántara, A., Sépulveda, E., Milford, C., González-Sicilia, P., and Torres, C.: New insights into the 2021 La Palma eruption degassing processes from direct-sun spectroscopic measurements, EGUsphere [preprint], https://doi.org/10.5194/egusphere-2025-1092, 2025.

Taylor, I. A., Preston, J., Carboni, E., Mather, T. A., Grainger, R. G., Theys, N., et al. (2018). Exploring the utility of IASI for monitoring volcanic $SO_2$ emissions. Journal of Geophysical Research: Atmospheres, 123, 5588–5606. https://doi.org/10.1002/2017JD027109.

**7. Terminology – The comparison of methods is presented largely in terms of institutions (i.e. IGN vs AEMET–ACTRIS). While this may be relevant for the authors, readers are likely to be more interested in the distinction between techniques, namely video-surveillance cameras versus ground-based profiling instruments. I would recommend revising the subscripts of the variable names accordingly.**

First, we would like to emphasize, as noted in the Specific Comment number 2, that the objective of this study is not to perform a conventional instrumental comparison (i.e., using identical instruments or approaches in order to obtain an unequivocal estimate of the error source responsible for the discrepancies observed in such an

analysis). Rather, the aim is to highlight the potential of combining or complementing different measurement techniques and methods to broaden the available information on the same physical phenomenon (volcanic plume height). For this reason, the authors consider it more appropriate to emphasize and acknowledge the origin of each database in relation to the technical and material efforts undertaken by both institutions (IGN and AEMET-ACTRIS).

We consider that, with this nomenclature and given the information presented in the manuscript, readers should have no doubts regarding the techniques deployed by IGN or AEMET-ACTRIS. Moreover, taking into account that AEMET-ACTRIS employs a multi-instrument and multi-approach methodology, the authors believe that making such a distinction based on the specific technique used would not necessarily provide further clarification.

**TECHNICAL REMARKS**

- Abstract – Please state why plume height determination is important already in the abstract.

  The following sentence has been included in the corrected manuscript:

  "These efforts are undertaken due to the importance of monitoring volcanic plume height in terms of air quality (necessary for the implementation of effective civil protection policies), volcanic activity surveillance (for tracking and forecasting eruptive behaviour), and, from a scientific perspective, for improving our understanding of the climatic and radiative impacts of this type of aerosol."

- Line 2 – Replace "volcanic event" with "eruption" on first mention > Done

- Lines 6–7 – The phrase "in collaboration" is repeated.

  This is the new sentence:

  "In parallel, the State Meteorological Agency of Spain (AEMET), in partnership with other Spanish ACTRIS (Aerosol, Clouds, and Trace Gases Research Infrastructure) members and collaborating institutions, conducted an unprecedented instrumental deployment to evaluate the impacts of this volcanic event on atmospheric composition."

- Line 17 – The phrase "different scientific perspective" is vague. Please clarify what the different perspective entails.

  This is the new sentence:

  "…the results of the present work provide complementary information and novel insights from **an alternative observational approach**, …"

- Lines 22–25 – Are $SO_2$ and the aerosol vertical profiles expected to be similar in proximity to the source? Please, explain the reason in the manuscript.

  In this line the authors state the importance of $h_{ec}$ for the $SO_2$ estimations done by using satellite observations. Satellite products that estimate $SO_2$ emissions during

volcanic eruptions are highly dependent on the layer height, which is a critical factor that often introduces uncertainty in their retrievals. Since the signal observed by the sensor depends on how $SO_2$ is distributed vertically in the atmosphere, the estimation of the layer height is a relevant input for the retrieval algorithm. The importance of having reliable information on the height of the eruptive column is highlighted in the text in several paragraphs, such as 73–82, in Section 3.3.4, and in paragraphs 545–550.

Regarding the specific question raised by this Referee, near the volcanic source, $SO_2$ and aerosol vertical profiles are expected to be roughly similar, especially before significant chemical conversion and particle settling occur. As explained in lines 216-219, given the proximity of the stations to the volcanic vent, no significant dispersive processes nor intra-plume chemical reactions are expected.

- Introduction – The section is rather wordy. Some paragraphs could be shortened, retaining only information directly relevant to the scope of the manuscript.

  This section has been changed according to Specific Comment number 1.

- Line 38 – The expression "conversion process from primary to secondary aerosols" is unclear. The preceding text refers to gas-to-aerosol conversion, not aerosol-to-aerosol processes. Please clarify.

  In this section of the introduction, the aim was to distinguish between sulfate aerosols emitted directly during the eruption (primarily emitted but secondary aerosols) and those formed later through conversion processes. In the text, we are referring to gas-to-aerosol conversion. However, this section of the introduction was summarized in accordance with Specific Comment 1, and that particular sentence was removed.

- Line 44 – "significantly longer": is this due to particle size, or to chemical composition? Please specify.

  In this paragraph, the authors intended to point out that the lifetime of sulfate aerosols is much longer than that of primary emissions, such as $SO_2$ or volcanic ash. This is due, on the one hand, to the reactivity of $SO_2$, which is rapidly converted into sulfate aerosol, and to the size and density of volcanic ash, which deposits quickly through sedimentation. Thus, it can be stated that the lifetime of sulfate aerosol is significantly longer due to both its size and chemical properties.

- Line 47 – Bibliographic references are required.

  A new reference to Zhu et al. (2020) has been added:

  Zhu, Yunqian ; Toon, Owen B. ; Jensen, Eric J. et al. / Persisting volcanic ash particles impact stratospheric $SO_2$ lifetime and aerosol optical properties. In: Nature Communications. 2020 ; Vol. 11, No. 1.

- Lines 83–91 – Given the large number of existing studies, the novelty of this work should be highlighted more clearly. The statement that "this study aims to present the unprecedented instrumental coverage..." is too generic. This would be the ideal

point to explicitly state the scientific questions that remain open and will be answered here.

This sentence has been changed according to Specific Comment number 1.

- Line 106 – Rather than emphasising the different networks, it may be more informative to highlight the differences between techniques, as noted above.

This question has been answered in Specific Comments number 1 and 2.

- Line 113 – Consider starting the sentence with "The eruption began..." > Done
- Lines 118–120 – As this is key information, it could also be included in the abstract > Done
- Lines 121–126 – These lines are only loosely connected to the paper's focus and could be condensed > Done

This is the new paragraph:

"The eruption caused long-lasting impacts due to extensive lava fields (1219 ha), widespread damage to infrastructure, homes, and farmland, and ongoing volcanic gas emissions in a region heavily dependent on tourism. Aviation was severely affected, with 26% of scheduled flights at La Palma Airport cancelled—34% due to airport closures from ash accumulation and 66% due to ash in the airspace. Initial estimates place total economic losses at over 1025 million USD Benito et al. (2023)."

- Lines 127–131 – This paragraph might be better placed earlier in the event description.

This paragraph is the first to properly describe the eruption in volcanological terms and is complemented by the information provided in the following paragraph, which defines the type of activity. At the beginning of the section, a general overview of the eruption is given (dates and societal impacts). The authors did not find an appropriate place at the beginning of the section to position the information from this paragraph.

- Section 3.1 – Please clarify whether the reported technique was developed specifically for this study or if it has been described previously.

Video surveillance technique is a widespread ground-based technique to monitor the altitude of the volcanic plume, as cited in the text. According to Felpeto et al. (2022), who described the specific network deployed in La Palma during the eruption, IGN used one pre-existing camera on Roque de los Muchachos Observatory, but they installed three more cameras, as described in the text. References have been added in the text to clarify that the technique is not new, but widely applied in the community. The existing ground-based techniques are also described in the introduction (lines 68-71).

- Figure 1 – It may be useful to explain why the grid is slightly tilted relative to the figure margins.

The slight tilt of the camera, detected through a specific calibration process, causes the altitude grid not to align perfectly with the edges of the image. Each line represents an elevation above sea level, but the entire grid appears tilted in the photograph due to the angle at which the camera was oriented.

- Figure 2 caption – "LA" → "La" > Done

- Line 191 – "in record time" is too informal. Instead, state the precise number of days.

  We have modified the text to state the number of days:

  "Four stations were strategically deployed in record time **(between 11 and 40 days after the eruption)** around the Tajogaite-Cumbre Vieja volcano by AEMET-ACTRIS members in Spain in collaboration with other institutions."

- Section 3.2 – Please homogenise the terminology for instruments (e.g. ceilometers vs lidar ceilometers) or explain why you keep the naming different. Note that ACTRIS currently uses "Automatic low-power lidars and ceilometers (ALCs)" or, more generally, "Automatic lidar ceilometers (ALCs)".

  Throughout most of the text, the authors generally refer to the profiler network and list the number of profilers and their characteristics, in order to avoid overly cumbersome terminology for such a diverse network. When describing each instrument in Section 3.2, we have defined the ARCADE as a Raman lidar, the MPL as a micro-pulse lidar, and the three remaining profilers (CHM15k, CL51, and CL61) as ceilometers. In our view, further subdividing these categories would be counterproductive for the reader's understanding. We have harmonized these terms also in lines 395, 523 and 530.

- Line 272 – As vertical resolution is an instrument characteristic, this should be mentioned directly after "backscattered signal" in the same sentence > Done

- Lines 274–277 – Were the profiles corrected for overlap? Please clarify.

  Yes, the overlap of this profiler was corrected according to the overlap function provided by the manufacturer. This information has been included in the corrected manuscript.

- Line 282 – "sharp gradients" – do you refer specifically to negative gradients? Please specify > Done

- Lines 292–293 – Why is this algorithm different? Was it validated against the others?

  The algorithm used for the CL51 and CL61 is the same, and in turn differs from that used for the three previous profilers. As noted in the Specific Comment 2, the consistency of the comparison and its usefulness for achieving the objective of this study does not rely on using identical instruments and methodologies.

- Line 292 – The "temporal evolution of attenuated backscatter profiles" does not appear to require retrievals. Are the authors instead referring to the identification of aerosol layer tops?

The authors agree with this comment. We have changed the sentence accordingly.

- Line 306 – Wording suggestion: "... pre-calibrated at the Vaisala facilities using liquid water clouds..." > Done

- Line 311 – Why are only six cases considered? Please explain.

  The authors have noted that it was not properly explained (in the appropriate section) how the AEMET database was constructed, nor the criteria by which, in cases where multiple profilers coincided with IGN observations, the height from one of them was given priority. For this reason, paragraphs 216–225 have been revised as presented below:

  "In cases where multiple AEMET heights from different profilers and sites coincide with the IGN height, the one most influenced by the volcanic plume, based on the prevailing wind direction, is selected."

  In this way, in response to the Referee question number 1, El Paso contributed only six measurements to the AEMET series because it was the last instrument to be installed and, even once it was operational, data from another profiler were considered more representative for inclusion in the final series.

- Lines 357–362 – As this mission ended on 1 August 2023, this paragraph should be written in the past tense > Done

- Line 364 – The term "loading" is ambiguous. Does it refer to emissions or concentrations? Similarly, "budgets" (line 384) is unclear. Lines 373–375 could be moved earlier to clarify the product description.

  We have changed loading and budget by $SO_2$ emissions. We have also moved lines 373-375 according to this comment.

- Section 4.1 – As noted above, it may be clearer to separate the discussion by technique rather than by research group.

  This comment has been addressed in the Specific Comment number 7.

- Figure 4 – Given the evident variability in plume height, short-term fluctuations (line 389) should be discussed in greater detail. To better assess the agreement/differences in altitude estimates, consider splitting Figure 4 into two or more panels covering different time intervals.

  The authors understand that the variability in plume height, inherently linked to the variability of volcanic activity, is high throughout most of the eruptive process. However, the definition in terms of the three eruptive phases is provided later and presented in Figure 5, corresponding to the RSAM signal (which is necessary for defining these phases). For this reason, adding subfigures at this point in the discussion using different time intervals (not yet defined) would not be advisable. Once the phases are defined, Figure 5 presents a cleaner series, divided by lines indicating the respective eruptive phase. We will rely on this combined figure, with the phases clearly defined, to describe the eruptive process.

- Line 396 – Please specify when the maximum difference occurred. Similarly, indicate when the minimum occurred (line 397).

  According to the General Comments from the Referee 2, a new table and a more detailed discussion on the results have been included in the corrected manuscript.

- Line 401 – "limited number of measurements": please restate the exact number > Done

- Line 482 – Is the volcanic plume consistently the highest aerosol layer?

  As is done with ground-based methodologies such as the Gradient Method, layer detection is performed by identifying gradients in the backscatter signal and, in the case of CALIOP, by also using the depolarization ratio along with other filters and algorithms that complement the classification. Clearly, it cannot be guaranteed that the volcanic aerosol layer is systematically the last layer measured by the profilers or the CALIOP sensor, but the product itself and the detected layer height provide guidance on whether it corresponds to the volcanic layer or to another type of aerosol that may be transported aloft. This was not the case for any of the observations conducted in the present study.

- Lines 597–600 – There appear to be typographical errors in this bibliography item.

  This error has been corrected.

---

## Author Comment (AC2)

**General Comments:**

This manuscript offers a thoroughly conducted and significant investigation into the characterization of plume heights during the 2021 Tajogaite (La Palma) eruption, utilizing complementary datasets from IGN video observations, AEMET–ACTRIS aerosol profilers, and satellite instruments. The study emphasizes the essential role of precise, real-time plume height measurements in ensuring reliable satellite retrievals of volcanic emissions and establishes a valuable framework for future volcanic crisis management.

The primary enhancements required are minor, including the incorporation of fundamental statistical indicators such as RMSE and correlation for the AEMET-IGN comparison (see Fig. 4), a concise sensitivity analysis of CALIOP results, clarification or correction within graphical regions, and the explicit delineation of uncertainty ranges.

The manuscript is clearly written, well organized, and enhanced with high-quality figures. It makes a significant and original contribution to atmospheric measurement science. I recommend acceptance after minor revisions.

**Answer from the authors to the general comments:** The authors appreciate the constructive comments provided by this referee. The first comment, concerning the inclusion of statistical analyses in the comparison of volcanic plume height databases, will be addressed in the specific responses, as well as in our reply to Referee 1. The comment regarding the analysis of the results of the comparison with CALIOP coincides with Specific Comment number 5 from the Referee 1. In this regard, a detailed explanation has been provided on the validity of this comparison, as well as on the classification of aerosol types performed by the CALIOP products for the events selected for the comparison. Finally, regarding the expected uncertainty of the CALIOP product, all references in the literature aim to assess its quality through comparison and validation strategies, such as those carried out in this work and cited in the corresponding section. Unfortunately, to the best of our knowledge, it is not possible to provide further information on the specific uncertainty ranges of this satellite product. A detailed response related to the possible effect of a wide overpass distance has been also given in the Referee 1 Specific Comment number 5.

**Technical Comments:**

**Regarding Figure 4: This plot would substantially benefit from the inclusion of a descriptive table that provides a comprehensive statistical analysis. This should include correlation coefficients between the AEMET-ACTRIS and IGN datasets for dispersive plume heights (h_d). Additionally, such a table could quantify differences stratified by AEMET instrument type in comparison to IGN**

**measurements, thereby enhancing the interpretability of inter-method consistency. Notably, multiple data points appear for the same day and source (particularly for h_d, IGN), which may introduce visual clutter; consolidating these into daily aggregates—such as means and standard deviations, where applicable—could improve clarity. Nonetheless, the table already presents a synthesized view, as it is effectively summarized in Figure 5a through daily averaged values by source. The proposed statistical table would thus serve as a valuable complement for a rigorous intercomparison.**

As stated in the response number 3 to the Referee 1 Scpecific Comments, we fully agree on the advantages of extending the discussion of the different techniques by adding relevant statistical information, as pointed out by the Referee in this comment. The details of this expanded discussion are presented below and are the same as those provided in response 3 to Referee 1.

[Line 402] **In light of the comparison results shown in Table 1, it can be observed that the methodology used to derive the volcanic plume height from lidar data does not appear to play a dominant role in the comparison outcomes, with the best statistics obtained for the FUE and TAZ cases (Gradient Method in both cases).**

Table 1: Main statistical skill scores (in m) for the comparison between $h_{d,IGN} - h_{d,AEMET}$ differences, including also the multi-instrument and inter-method comparison between $h_{d,IGN}$ and the height of the volcanic plume measured by AEMET at the five different stations. The methodology for retrieving the altitude of the volcanic plume is also included. GM stands for the Gradient Method, and WCT for the Wavelet Covariance Transfer method.

| | $h_{d,IGN} - h_{d,AEMET}$ | $h_{d,IGN} - h_{d,ORM}$ | $h_{d,IGN} - h_{d,FUE}$ | $h_{d,IGN} - h_{d,AER}$ | $h_{d,IGN} - h_{d,TAZ}$ | $h_{d,IGN} - h_{d,EP}$ |
|---|---|---|---|---|---|---|
| Methodology | Multi-approach | Qualitative | GM | WCT | GM | WCT |
| Mean Difference | 258.6 | -139.8 | 203.6 | 430.4 | 344.8 | 941.7 |
| Standard Deviation | 620.4 | 866.5 | 531.7 | 540.7 | 487.1 | 468.5 |
| RMSE | 672.1 | 877.7 | 569.4 | 691.1 | 596.8 | 1051.8 |
| Pearson coefficient (r) | 0.81 | 0.18 | 0.86 | 0.64 | 0.89 | 0.72 |
| Slope | 0.90 | 0.10 | 0.86 | 0.99 | 0.99 | 1.16 |
| Intercept | 103.4 | 3918.8 | 307.1 | -426.4 | -302.9 | -1520.6 |

**Regarding Figure 6: The discussion of wind direction analysis in lines 473–479 lacks sufficient clarity, particularly in elucidating the methodological basis for the wind rose construction. Intermediate directional sectors (intercardinal headings) between principal cardinal points (N–E–S–W) are incorrectly labeled; for instance, the sector between south and west should be designated SW, with analogous corrections for other quadrants. The boundary between W and SW, corresponding to 247.5° (referenced to 0° as north) as WSW, exemplifies this issue. Conventionally, wind roses depict the direction *from* which the wind originates (provenance), rather than toward which it blows. I recommend redrawing the wind rose with explicit labeling of the directional convention (e.g., "wind from" or "wind toward") to avoid ambiguity. Furthermore, while Figure 1 accurately positions the Tazacorte (west) and Fuencaliente (south) stations relative to the island, the text and**

**Figure 6 introduce confusion in their spatial referencing, which should be reconciled for consistency.**

The authors fully agree with this comment. We appreciate this observation regarding the issues in the figure, as it has helped us improve the presentation of the results and avoid potential confusion for the reader. Indeed, there was a clear error in the labels corresponding to the half-axes of the cardinal points, which has now been corrected. Additionally, a new plot has been produced, this time using bars, to allow a clearer analysis of wind direction. This analysis has been conducted taking into account the origin of the wind to avoid any ambiguity, as the referee states in his/her comment.

This is the new corrected figure and figure caption:

[Figure]

Figure 6. Wind rose diagram for the volcanic eruptive column hec (in m a.s.l.) and HARMONIE-AROME wind **direction** at the pixel **above** the volcanic edifice in Cumbre Vieja. **Wind direction is given following the meteorological convention (direction of origin).**

The new text in the corrected manuscript is the following (lines 478-479):

"This distribution highlights the role of wind direction in plume transport and vertical dispersion, further evidenced in Fig. 6, which displays the wind rose diagram for the study period. **The trade wind regime (originating from the NE) remains predominant while the volcanic plume height stays within the marine boundary layer (up to 2000 m a.s.l.). Above this level, a change in prevailing wind direction occurs, as expected from the vertical balance of forces at this latitude. Up to 4000 m a.s.l., two predominant components are observed (NE and W–WSW). It must be noted that this analysis reflects the wind structure under the specific meteorological conditions of the 85-day eruption period and cannot be interpreted as a climatological pattern.**"

**Regarding Figure 8 and section 4.4: the data points representing the eruptive column height (h_ec) are indicated by blue circles, not orange as might be inferred from the caption or legend—please verify and rectify this for accuracy. Additionally, the use of the color "red" may be preferable to "light red." The SO$_2$ emission rates are expressed in kilotonnes, yet they seem to pertain to daily fluxes (kt day⁻¹); explicitly stating the temporal averaging (e.g., daily emission rates [kt·day⁻¹]) in the axis labels and accompanying text would prevent misinterpretation. Furthermore, it is advisable to review the bibliography to include any prior studies that report similar underestimations of satellite-derived emission rates during volcanic eruptions (e.g., via UV hyperspectral retrievals). If applicable, incorporate references to complementary ground-based or alternative methodological estimates of SO$_2$ emission rates (kt day⁻¹), such as differential optical absorption spectroscopy (DOAS) or flux tower measurements related to Tajogaite, in order to provide a more comprehensive contextualization of the findings.**

The authors fully agree with this comment, which has allowed us to correct the error in the color attribution in Figure 8, as well as the typographical error related to the units of the SO$_2$ emission fluxes.

This is the new corrected figure and the figure caption:

[Figure]

Figure 8. NASA **daily** SO$_2$ **emission** (in kilotons **per day**, kt **day$^{-1}$**) calculated as the average of OMI, OMPS and TROPOMI UV backscatter radiances considering a standard columnar injection height of 8 km (**blue circles**) and the real h$_{ec}$ measured by the IGN (in **red triangles**). Dotted vertical lines represent the three eruptive phases derived from RSAM data series.

Regarding the new references to be added in the text, the new paragraph is the following:

"Consistent daily emissions for the Tajogaite volcano were calculated by Esse et al. (2025) by means of a novel forward trajectory approach using PlumeTraj, a pixel-based trajectory analysis of an SO$_2$ cloud Pardini et al. (2017). Ground-based miniDOAS observations of SO$_2$ emissions were also carried out during the Tajogaite eruption. This is exemplified by the studies of Albertos et al. (2022) and Rodriguez et al. (2023), which reported SO$_2$ emissions ranging from 670 to 17 tons per day and observed a clear decreasing trend in SO$_2$ during the post-eruptive phase."

The authors have also added new reference to $SO_2$ emissions published by Esse et al. (2025) as a result of the Specific Comment number 6 of the Referee 1. The following information has been included in the corrected manuscript (line 506):

"Comparable daily $SO_2$ emission values to those retrieved using $h_{ec,IGN}$ were reported by Esse et al. (2025) for the same eruption, based on the PlumeTraj analysis, with lower mean relative differences of 21.6%."

References:

Pardini F, Burton M, Vitturi MD, Corradini S, Salerno G, Merucci L, Di Grazia G (2017) Retrieval and intercomparison of volcanic $SO_2$ injection height and eruption time from satellite maps and groundbased observations. J Volcanol Geoth Res 331:79–91.

Albertos, V. T., Recio, G., Alonso, M., Amonte, C., Rodríguez, F., Rodríguez, C., Pitti, L., Leal, V., Cervigón, G., González, J., Przeor, M., Santana-León, J. M., Barrancos, J., Hernández, P. A., Padilla, G. D., Melián, G. V., Padrón, E., Asensio-Ramos, M., and Pérez, N. M.: Sulphur dioxide ($SO_2$) emissions by means of miniDOAS measurements during the 2021 eruption of Cumbre Vieja volcano, La Palma, Canary Islands, EGU General Assembly 2022, Vienna, Austria, 23–27 May 2022, EGU22-5603, https://doi.org/10.5194/egusphere-egu22-5603, 2022.

Rodríguez, O., Barrancos, J., Cutillas, J., Ortega, V., Hernández, P. A., Cabrera, I., and Pérez, N. M.: $SO_2$ emissions during the post-eruptive phase of the Tajogaite eruption (La Palma, Canary Islands) by means of ground-based miniDOAS measurements in transverse mode using a car and UAV, EGU General Assembly 2023, Vienna, Austria, 23–28 Apr 2023, EGU23-3620, https://doi.org/10.5194/egusphere-egu23-3620, 2023.

Esse B, Burton M, Hayer C, La Spina G, Pardo Cofrades A, Asensio-Ramos M, Barrancos J, Pérez N. Forecasting the evolution of the 2021 Tajogaite eruption, La Palma, with TROPOMI/PlumeTraj-derived $SO_2$ emission rates. Bull Volcanol. 2025;87(3):20. doi: 10.1007/s00445-025-01803-6. Epub 2025 Feb 26. PMID: 40028348; PMCID: PMC11865176.